# GEMQ: Global Expert-Level Mixed-Precision Quantization for MoE LLMs

**Jianing Deng** [1] **Song Wang** [2] **Dongwei Wang** [3] **Zijie Liu** [4] **Tianlong Chen** [4] **Huanrui Yang** [3] **Jingtong Hu** [1]

## Abstract

Mixture-of-Experts Large Language Models (MoE-LLMs) achieve strong performance but incur substantial memory overhead due to massive expert parameters. Mixed-precision quantization mitigates this cost by allocating expert-wise bit-widths based on their importance, approaching the accuracy-memory Pareto frontier and enabling extreme low-bit quantization. However, existing methods rely on layer-wise importance estimation and overlook router shifts induced by quantization, resulting in suboptimal allocation and routing. In this work, we propose **G**lobal **E**xpert-level **M**ixed-precision **Q**uantization (**GEMQ**) to overcome these limitations via (1) a global linear-programming formulation that captures model-wide expert importance based on quantization error analysis, and (2) efficient router fine-tuning to adapt routing to quantized experts. These components are integrated into a progressive quantization framework that iteratively refines importance estimation and allocation. Experiments demonstrate that GEMQ significantly reduces memory and accelerates inference with minimal accuracy degradation. Source code is available at: https://github.com/jndeng/GEMQ.

## 1. Introduction

Large Language Models (LLMs) have achieved state-of-the-art performance on a wide range of natural language processing tasks (Minaee et al., 2024). Among recent advancements, Mixture-of-Experts (MoE) architectures have emerged as a prominent scaling paradigm (Shazeer et al., 2017; Muennighoff et al., 2024; Jiang et al., 2024; Dai et al., 2024; Yang et al., 2025). MoE models utilize a sparse architecture wherein a routing mechanism selectively activates a small subset of expert networks for each input. This design facilitates efficient scaling and enables MoE-LLMs to match the performance of dense counterparts with a fraction of the computational cost. However, while this approach reduces the computational load per input, the total number of model parameters remains unchanged. All experts must be co-located in memory during inference, resulting in a substantial memory footprint that presents a significant deployment challenge (Kim et al., 2023b; Li et al., 2024; Huang et al., 2024a). Even high-end GPUs such as the NVIDIA H100-80GB are insufficient to accommodate typical MoE models like Mixtral-8×7B (Jiang et al., 2024) in full-precision. Therefore, effective model compression is critical for practical deployment of MoE-LLMs.

Quantization, which reduces the numerical precision of model parameters, has emerged as a promising compression technique widely applied to dense LLMs (Wan et al., 2023; Frantar et al., 2022; Lin et al., 2024; Wang et al., 2026). However, naively applying these methods to MoE-LLMs overlooks the unique properties of MoE architectures and leads to severe performance degradation (Li et al., 2024). On one hand, the primary objective of MoE-LLMs quantization is to reduce the size of expert parameters, which typically account for over 90% of the total model parameters and dominate memory consumption (Kim et al., 2023b; Huang et al., 2024a). On the other hand, recent studies (Lu et al., 2024; Huang et al., 2024a) have shown that experts exhibit distinct activation patterns, indicating that not all experts are equally important. This motivates expert-level mixed-precision quantization for MoE-LLMs (Li et al., 2024; Huang et al., 2024a), where experts are assigned different bit-widths according to their relative importance. Specifically, expert bit allocation can be formulated as a linear programming (LP) problem, with statistics (*e.g.*, expert activation frequency) measured on the full-precision (FP) model serving as proxies for importance. Once the LP determines the bit assignment, a post-training quantization algorithm such as GPTQ (Frantar et al., 2022) is applied to each expert accordingly. This mixed-precision strategy substantially outperforms uniform allocation, preserving high performance even under extremely low-bit settings (Huang et al., 2024a).

Despite their effectiveness, we identify two primary limitations in existing mixed-precision quantization methods for

[1]University of Pittsburgh [2]University of Central Florida [3]University of Arizona [4]University of North Carolina at Chapel Hill. Correspondence to: Jianing Deng <jid70@pitt.edu>, Jingtong Hu <jthu@pitt.edu>.

*Proceedings of the $43^{rd}$ International Conference on Machine Learning*, Seoul, South Korea. PMLR 306, 2026. Copyright 2026 by the author(s).

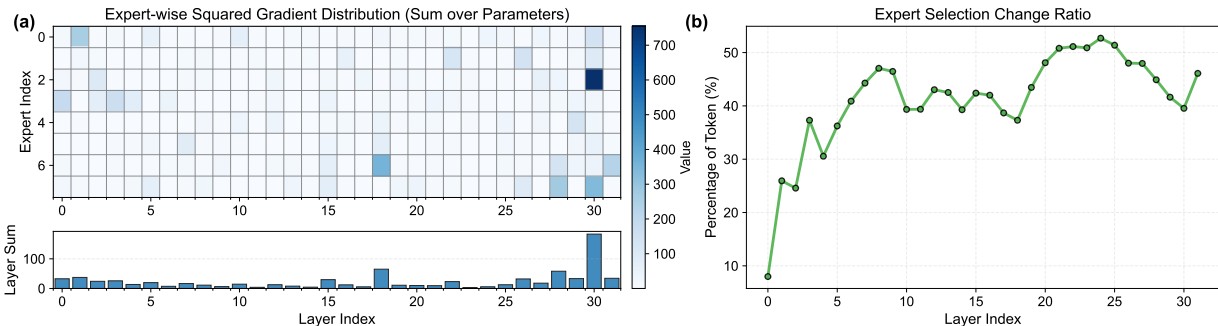

*Figure 1.* **Motivations.** (a) The sensitivity of expert weights, measured by squared gradients (*i.e.*, the trace of the empirical Fisher Information Matrix), varies not only within a layer but also across layers, indicating heterogeneous layer importance; and (b) over 40% of tokens are routed to different experts after 1.5-bit quantization, revealing substantial distortion in router distributions. Statistics are computed from the Mixtral-8×7B model on WikiText2.

MoE-LLMs: **(1)** bit allocation is performed locally at the layer level, considering only the relative importance of experts within each layer and assigning an identical bit budget to every layer. However, as shown in Fig. 1 (a), we observe that different layers exhibit varying levels of importance (sensitivity) and thus necessitate different bit budgets; and **(2)** as shown in Fig. 1 (b), low-bit quantization substantially alters the dynamics of MoE routers, leading to large shifts in token-to-expert assignments. However, existing work largely overlooks this discrepancy, resulting in suboptimal routing and degraded performance in quantized models.

To address these issues, we propose Global Expert-level Mixed-precision Quantization (GEMQ), which enables effective quantization for MoE-LLMs under extreme low-bit settings. Our method addresses the aforementioned issues in both the pre-quantization and post-quantization phases. In pre-quantization, we formulate a linear programming model for expert-level bit-width allocation from an error-analysis perspective, where each expert's importance is derived globally based on the task loss. This formulation allows simultaneous consideration of expert importance across the model and naturally extends the layer-wise local LP (Huang et al., 2024a) into a global LP. In post-quantization, we propose a simple yet effective approach that adapts routers to quantized experts by fine-tuning router weights, which constitute less than 0.04% of the total parameters. This parameter-efficient fine-tuning (PEFT) strategy requires only a small calibration set and induces less than 5% time overhead in the quantization process, while enabling optimal expert selection and substantially improving quantization performance. Additionally, we analyze the loss landscape of MoE-LLMs and propose integrating the two techniques into a progressive quantization framework to alleviate inaccurate expert importance estimation under large quantization errors. Specifically, instead of using the FP model for bit allocation, we leverage the quantized and fine-tuned model under the previous bit budget, thereby mitigating the FP-quantized discrepancy and yielding more accurate expert bit-width allocation.

We conduct extensive experiments on diverse MoE-LLMs and benchmarks, demonstrating that GEMQ substantially reduces memory usage while largely preserving model capabilities. Specifically, when quantizing the 16-bit Mixtral-8×7B (Jiang et al., 2024) model to 2.5 bits per expert, our method reduces the model size from 87 GB to 16 GB (an 82% reduction), with only a 7% accuracy drop on 5-shot MMLU (Hendrycks et al., 2020), outperforming the state-of-the-art mixed-precision quantization method PMQ (Huang et al., 2024a) by about 3%. Using our optimized MoE kernel implementation, the 2.5-bit quantized Mixtral-8×7B model can be deployed on a single NVIDIA H100 GPU, achieving a decoding speed of 82.5 tokens per second.

## 2. Related Work

**Quantization for LLMs.** Post-training quantization (PTQ) has become a widely adopted compression technique for dense LLMs, as it requires no additional training (Dettmers et al., 2022; Frantal et al., 2022; Xiao et al., 2023; Shao et al., 2023; Kim et al., 2023a; Lin et al., 2024; Tseng et al., 2024; Ashkboos et al., 2024; Sun et al., 2025). Quantization-aware training (QAT) is more effective in preserving the performance of quantized lightweight LLMs, though it requires substantial retraining resources (Liu et al., 2024b; Chen et al., 2025a; Liu et al., 2025). Our method integrates PTQ with parameter-efficient fine-tuning (PEFT), achieving improved performance while incurring minimal overhead. Since the primary bottleneck for generative inference with LLMs is memory bandwidth (Kim et al., 2023a; Lin et al., 2024), many studies focus on weight-only quantization (Frantar et al., 2022; Xiao et al., 2023; Kim et al., 2023a; Lin et al., 2024), where model weights are quantized to low bits while computations are performed in full precision. In this work, we also focus on weight-only quantization, as the deployment challenges of MoE models stem primarily from substantial memory pressure. On the other hand, prior work has explored the varying sensitivity of weights in LLMs and proposed mixed-precision quantization methods that allocate different bit widths accordingly (Dong et al.,

2020; Li et al., 2021; Dettmers et al., 2022; 2023; Huang et al., 2024c;b). However, applying these techniques to MoE-LLMs remains challenging due to the unique expert routing mechanism in MoE architectures.

**Quantization for Mixture-of-Experts LLMs.** Early studies (Kim et al., 2023b; Frantar & Alistarh, 2023) on MoE-LLMs quantization assign a uniform bit-width to all experts and directly apply PTQ methods (Frantar et al., 2022; Lin et al., 2024) developed for dense LLMs for quantization. However, such approaches overlook the sparsity inherent to the MoE architecture, leading to suboptimal performance. Li et al. (2024) pioneer the study of expert-level mixed-precision quantization, proposing a bit allocation strategy based on expert activation frequency. PMQ (Huang et al., 2024a) further advances this line of work by formulating expert bit-width allocation as a linear programming (LP) problem and employing a heuristic coefficient that combines expert activation frequency with weight statistics to determine expert importance. Building upon this, Duanmu et al. (2025) explore finer-grained sub-expert bit allocation and incorporate hardware-aware co-design into the LP formulation. However, existing approaches remain constrained to local, layer-wise bit allocation and thus fail to capture variations in expert importance across layers. Beyond mixed-precision allocation, other efforts address different aspects of MoE-LLMs quantization, including calibration dataset construction (He et al., 2026; Hu et al., 2025; Zheng et al., 2025) and mitigating router distribution shifts caused by expert quantization (Chen et al., 2025b; Fu et al., 2025). However, existing approaches for handling router distribution shifts rigidly enforce alignment with the full-precision router distribution, limiting adaptability and proving suboptimal for mixed-precision MoE-LLMs quantization in our experiments.

## 3. Preliminaries

**Mixture-of-Experts.** MoE-LLMs replace conventional feed-forward networks (FFNs) with MoE blocks, each comprising a router FFN and $N$ expert FFNs (Gale et al., 2023). For each input token $\mathbf{x}$, the router first computes routing logits $\mathbf{r} = \{r_0, r_1, \ldots, r_{N-1}\}$ and scores $\mathbf{s} = \mathrm{Softmax}(\mathbf{r})$. The top-$K$ ($K \ll N$) experts with the highest scores are then selected, and their outputs are aggregated through a weighted sum to produce the output $\mathbf{z}$ of the MoE block:

$$\mathbf{z} = \sum_{i=0}^{K-1} \frac{\mathbf{s}_i}{\sum_{j=0}^{K-1} \mathbf{s}_j} \, \mathrm{E}_i(\mathbf{x}) \,, \tag{1}$$

where $\mathrm{E}_i$ denotes the feed-forward operator of the $i$-th expert FFN.

**LLM Quantization.** Quantization maps floating-point weights from the range $[\mathbf{w}_{\min}, \mathbf{w}_{\max}]$ to integers in

$[0, 1, \ldots, 2^b - 1]$, where $b$ denotes the target bit-width. For LLM quantization, the primary objective is to quantize linear layers, which account for the majority of model parameters. Among existing methods, GPTQ (Frantar et al., 2022) is one of the most widely adopted approaches for quantizing linear layers in dense LLMs. It determines the optimal quantization mapping for each linear layer by minimizing the reconstruction error on a small calibration dataset:

$$\arg\min{}_{\hat{\mathbf{w}}} \quad \|\mathbf{w}\mathbf{x} - \hat{\mathbf{w}}\mathbf{x}\|_2^2 \,, \tag{2}$$

where $\hat{\mathbf{w}}$ denotes the quantized weights of the linear layer. Leveraging Hessian-based estimation ($\mathbf{H} = 2\mathbf{x}\mathbf{x}^\top$) and lazy batch updates, GPTQ can efficiently quantize Mixtral-8×7B (47B parameters) in 40 minutes on a NVIDIA H100 GPU. In this work, we adopt GPTQ as the underlying quantizer.

**Analysis of Quantization Error.** Consider a block with weights $\mathbf{w}$ whose output $\mathbf{z}$ is given by $\mathbf{z} = \mathcal{F}_{\mathbf{w}}(\mathbf{x})$. Quantization can be viewed as a weight perturbation that introduces noise $\Delta\mathbf{w}$ to the pretrained weights $\mathbf{w}$, yielding quantized weights $\hat{\mathbf{w}} = \mathbf{w} + \Delta\mathbf{w}$. To quantitatively analyze the resulting loss degradation, Nagel et al. (2020) employ a Taylor series expansion to approximate the loss increase $\Delta\mathcal{L}$ as:

$$\mathbb{E}[\mathcal{L}(\hat{\mathbf{w}}) - \mathcal{L}(\mathbf{w})] \approx \Delta\mathbf{w}^\top \mathbf{g}^{(\mathbf{w})} + \tfrac{1}{2}\Delta\mathbf{w}^\top \mathbf{H}^{(\mathbf{w})}\Delta\mathbf{w} \,, \tag{3}$$

where $\mathbf{g}^{(\mathbf{w})} = \mathbb{E}[\nabla_{\mathbf{w}}\mathcal{L}]$ and $\mathbf{H}^{(\mathbf{w})} = \mathbb{E}[\nabla_{\mathbf{w}}^2\mathcal{L}]$ denote the gradient and Hessian w.r.t. the block weights.

Due to the massive number of parameters in LLMs, explicitly computing the Hessian $\mathbf{H}^{(\mathbf{w})}$ is prohibitive. Therefore, Li et al. (2021) propose transforming the second-order error into the block outputs. Specifically, assuming the pre-trained model has converged to a local minimum, the gradient term in Eq. 3 is near zero and can be neglected, and the Hessian can be approximated by the Gauss-Newton matrix $\mathbf{H}^{(\mathbf{w})} \approx \mathbf{G}^{(\mathbf{w})} = \mathbf{J}_{(\mathbf{z})}(\mathbf{w})^\top \mathbf{H}^{(\mathbf{z})}\mathbf{J}_{(\mathbf{z})}(\mathbf{w})$ (Botev et al., 2017), where $\mathbf{J}_{(\mathbf{z})}(\mathbf{w})$ is the Jacobian of the block outputs w.r.t. the block weights. Applying this approximation, Eq. 3 can be obtained as $\Delta\mathcal{L} \approx \tfrac{1}{2}\left(\Delta\mathbf{w}\mathbf{J}_{(\mathbf{z})}(\mathbf{w})\right)^\top \mathbf{H}^{(\mathbf{z})}\left(\Delta\mathbf{w}\mathbf{J}_{(\mathbf{z})}(\mathbf{w})\right)$. Li et al. (2021) further show that the perturbation-Jacobian product admits a first-order Taylor approximation of the block output change, *i.e.*, $\Delta\mathbf{z} \approx \Delta\mathbf{w}\mathbf{J}_{(\mathbf{z})}(\mathbf{w})$. Eq. 3 then becomes:

$$\mathbb{E}[\mathcal{L}(\hat{\mathbf{w}}) - \mathcal{L}(\mathbf{w})] \approx \tfrac{1}{2}\Delta\mathbf{z}^\top \mathbf{H}^{(\mathbf{z})}\Delta\mathbf{z} \,, \tag{4}$$

where $\Delta\mathbf{z} = \hat{\mathbf{z}} - \mathbf{z}$, and $\hat{\mathbf{z}} = \mathcal{F}_{\hat{\mathbf{w}}}(\mathbf{x})$ denotes the block outputs perturbed by the quantization error $\Delta\mathbf{w}$. In practice, due to the high cost of computing the Hessian $\mathbf{H}^{(\mathbf{z})}$, it is commonly approximated by the diagonal Fisher Information Matrix (FIM) (Li et al., 2021; Kim et al., 2023a) as:

$$\mathbf{H}^{(\mathbf{z})} \approx \mathrm{diag}\left(\mathbf{g}^{(\mathbf{z})}\mathbf{g}^{(\mathbf{z})\top}\right) \,, \tag{5}$$

where $\mathbf{g}^{(\mathbf{z})}$ denotes the gradient of the block outputs $\mathbf{z}$.

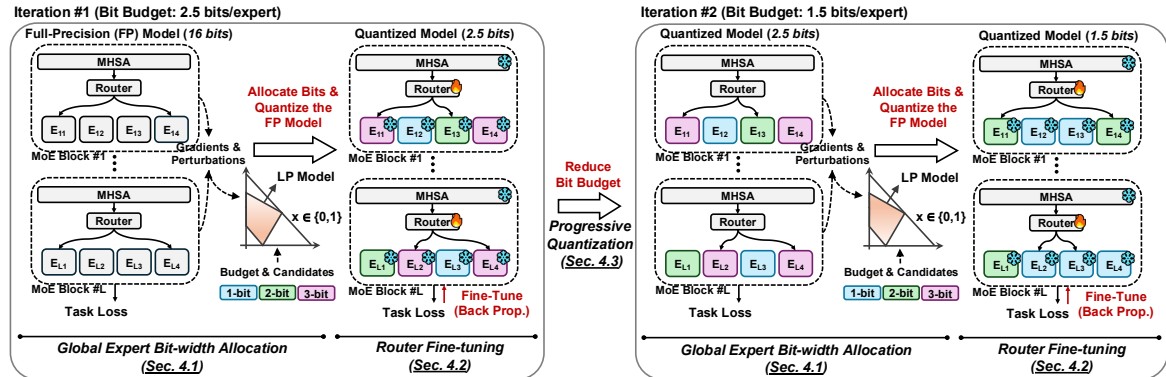

*Figure 2.* Overview of the proposed GEMQ framework for MoE-LLMs quantization.

## 4. Method

Fig. 2 presents an overview of the proposed GEMQ framework for MoE-LLMs quantization, which operates progressively. In each iteration under a given bit budget, we first apply a global allocation method to determine the optimal bit-width for each expert (Sec. 4.1). We then introduce an efficient global router fine-tuning method to solve the suboptimal router selection problem caused by expert quantization (Sec. 4.2). Finally, in Sec. 4.3, we integrate these two techniques into a progressive quantization framework to improve expert importance estimation under low-bit settings.

### 4.1. Global Expert Bit-width Allocation

Given a set of candidate bit-widths $\mathcal{B}$ (*e.g.*, $\mathcal{B} = \{1, 2, 3\}$) and a total bit budget $B$, the objective of expert-level bit-width allocation is to assign a bit-width to each expert $\mathrm{E}_i$ such that the sum of all assigned bits does not exceed $B$, while minimizing the increase in task loss $\mathcal{L}_{\mathrm{CE}}$ (*i.e.*, cross-entropy for language modeling) incurred by quantization.

To estimate how quantizing an individual expert FFN affects task loss, we apply Eq. 4 at the expert level using a diagonal FIM approximation of the output Hessian. Specifically, the increase in task loss caused by quantizing the $i$-th expert to $j$-bit, $j \in \mathcal{B}$, can be approximated as:

$$
\begin{aligned}
\Delta \widetilde{\mathcal{L}}_{ij} &\triangleq \mathbb{E}_{\mathcal{D}} \left[ \Delta \mathbf{z}_{ij}^{\top} \operatorname{diag}\left( \mathbf{g}_i^{(\mathbf{z})} \mathbf{g}_i^{(\mathbf{z})\top} \right) \Delta \mathbf{z}_{ij} \right] \\
&\approx \mathbb{E}_{\mathcal{D}} \left[ \mathcal{L}_{\mathrm{CE}}(\hat{\mathbf{w}}_{ij}) - \mathcal{L}_{\mathrm{CE}}(\mathbf{w}_i) \right],
\end{aligned}
\tag{6}
$$

where $\mathbb{E}_{\mathcal{D}}$ denotes the expectation over a calibration dataset $\mathcal{D}$, $\hat{\mathbf{w}}_{ij} = \mathbf{w}_i + \Delta \mathbf{w}_{ij}$ denotes the quantized expert weights, $\Delta \mathbf{z}_{ij} = \hat{\mathbf{z}}_{ij} - \mathbf{z}_i$ denotes the perturbation of the MoE layer output induced by quantization, and $\mathbf{g}_i^{(\mathbf{z})}$ denotes the gradients with respect to the corresponding layer outputs. Notably, $\mathbf{z}$ represents the aggregated MoE layer output rather than individual expert FFN outputs, thereby incorporating routing scores and properly weighting each expert's contribution by its routing probability.

Intuitively, $\Delta \widetilde{\mathcal{L}}_{ij}$ approximates the increase in task loss as the combined effect of local expert perturbations and the sensitivity of the corresponding layer. Since Eq. 6 is defined over the same task loss for all experts in the model, it enables direct comparison across experts, unlike prior layer-wise reconstruction objectives (Huang et al., 2024a; Duanmu et al., 2025).

With Eq. 6, we obtain a proxy of expert importance for bit allocation: if quantizing an expert leads to a substantial increase in loss, that expert is considered important and should be allocated a higher bit-width. To determine the optimal bit-width for each expert based on its importance, we formulate the following binary linear programming (LP) problem, which jointly considers all experts in the model and can be solved within seconds:

$$
\begin{aligned}
&\min_{\{x_{ij}\}} \quad \sum_{i \in \mathcal{E}} \sum_{j \in \mathcal{B}} \Delta \widetilde{\mathcal{L}}_{ij} \cdot x_{ij} \\
&\text{s.t.} \quad x_{ij} \in \{0, 1\}, \ \sum_{i,j} j \cdot x_{ij} \leq B, \ \sum_{j} x_{ij} = 1,
\end{aligned}
\tag{7}
$$

where $x_{ij}$ is a binary decision variable indicating whether the $i$-th expert is assigned a $j$-bit quantization, and $\mathcal{E}$ denotes the set of all experts in the model. Additionally, we impose a constraint requiring each layer to include at least one high-bit expert, which empirically serves as a mild regularizer and mitigates inaccurate importance estimation under low-bit settings. Unlike existing methods that rely on heuristic coefficients with hand-tuned hyperparameters for estimating expert importance, our formulation is hyperparameter-free and readily adaptable to different MoE-LLMs. After determining the optimal bit-width assignment for each expert in the model, we apply the GPTQ algorithm for quantization.

### 4.2. Global Router Fine-tuning

Due to the unique gating mechanism of MoE, quantizing experts can substantially affect router dynamics, altering both the router's input distribution and the behavior of routed experts, which in turn changes token-to-expert assignments,

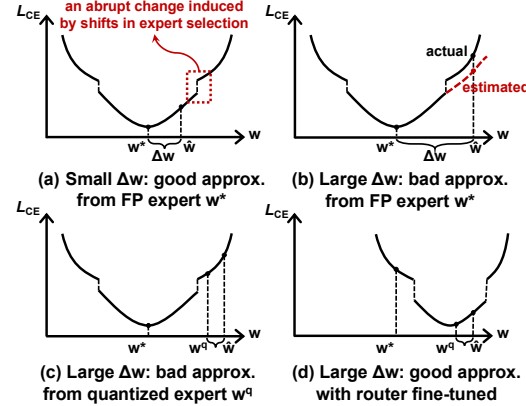

*Figure 3.* Illustration of different cases of expert importance estimation for target weights $\hat{\mathbf{w}}$. $\mathbf{w}^\star$ denotes FP expert weights.

as shown in Fig. 1 (b). Prior work on MoE-LLMs quantization overlook this issue, leading to suboptimal expert selection and performance degradation after quantization. To address this issue, we propose calibrating routers holistically by jointly tuning all router parameters after expert quantization to adjust the global dynamics. As shown in Fig. 2, we initialize the model with quantized weights (dequantized to floating-points in practice), freeze the attention and expert weights, and update only the router weights to minimize the cross-entropy task loss on a calibration set. The training process then guides the routers toward optimal routing decisions for expert selection. As shown in Tab. 8, since routers in MoE-LLMs typically constitute less than 0.04% of the total model parameters, the fine-tuning is inherently parameter-efficient and can be performed with minimal resources. For instance, fine-tuning the Mixtral-8×7B model on a calibration set of 128 sequences (2048 tokens each) for one epoch takes less than one minutes on three NVIDIA H100 GPUs, accounting for only 3.5% of the GPTQ quantization time. Yet this simple fine-tuning yields significant performance gains for quantized models, particularly in low-bit scenarios where router distributions change substantially, as evidenced in Sec. 5.

### 4.3. Progressive Quantization

In previous sections, we assume a valid Taylor series expansion of Eq. 3 for loss approximation when estimating expert importance. However, due to the unique loss landscape of MoE models, this approximation can become unreliable under low-bit quantization, where the quantization error $\Delta \mathbf{w}$ is large. In Fig. 3, we provide a 1-D illustration of the loss landscape of an expert in MoE-LLMs. As shown, perturbations to the weight $\mathbf{w}$ can cause abrupt changes in loss when they alter subsequent router selections. As shown in case (b), for large $\Delta \mathbf{w}$, the Taylor approximation around full-precision (FP) weights becomes inaccurate for estimat-

ing the loss of target weights $\hat{\mathbf{w}}$, due to abrupt loss changes induced by shifts in subsequent router selection. On the other hand, using weights closer to the target for estimation can alleviate the inaccuracy caused by abrupt changes, as shown in case (c). We hypothesize that a quantized model with a bit budget close to the target model can serve as such a good estimator. Nevertheless, using a quantized model remains problematic, as the quantized expert is no longer optimal and thus violates the local minimum assumption, rendering Eq. 4 inaccurate. As shown in case (d), using a fine-tuned model resolves this issue, as router fine-tuning ensures optimal expert selection and brings the quantized weights closer to the local minimum.

Based on the analysis, we propose a progressive quantization strategy that leverages a fine-tuned model with a close bit budget to improve expert importance estimation accuracy. Specifically, let $K$ denote the number of target bit budgets, sorted in *descending order* within $[B_1, B_K]$. We begin by quantizing the model at the highest bit budget $B_1$, where the FP model is used to extract the LP coefficients in Eq. 6, followed by router fine-tuning to obtain the quantized model $\mathbf{Q}_{B_1}$. Since the quantization error $\Delta \mathbf{w}$ remains small at high bit-widths, the estimation in Eq. 6 is reliable and $\mathbf{Q}_{B_1}$ achieves strong performance. We then progressively reduce the bit budget. For the $k$-th budget ($k > 1$), instead of reusing the FP model to compute Eq. 6, we leverage the previously fine-tuned model $\mathbf{Q}_{B_{k-1}}$. This design yields more accurate expert importance estimation in low-bit scenarios and improves quantization performance. The full algorithm is provided in Appendix F (Algorithm 1).

## 5. Experiments

**Models and Datasets.** We evaluate our method on four MoE-LLMs with varying characteristics: DeepseekV2-Lite (Dai et al., 2024), Qwen1.5-MoE-A2.7B (Yang et al., 2025), Qwen3-30B-A3B (Yang et al., 2025), and Mixtral-8×7B (Jiang et al., 2024). For evaluation, we report perplexity on the general-domain datasets WikiText2 (Merity et al., 2016) and C4 (Raffel et al., 2020) to measure token prediction accuracy. We further evaluate quantized LLMs on seven commonsense and reasoning benchmarks via the EleutherAI LM Harness (Gao et al., 2024). Further details of the models, datasets, and evaluation protocol are provided in Appendix A.

**Implementation Details.** For bit-width allocation, we follow Huang et al. (2024a) and use the general-domain text dataset C4 for gradient extraction, sampling 128 random sequences of 2048 tokens each. For quantization, we follow prior work (Li et al., 2024; Huang et al., 2024a) to quantize all attention modules to 4 bits and assigning each expert a bit-width from $\{1, 2, 3\}$ according to our allocation results.

*Table 1.* Comparison of perplexity on the WikiText2 (WT2) and C4 test sets, and average accuracy 0-shot$_7$ (%) across seven zero-shot tasks. #Bits denotes the average bit-width per expert. The best results in each comparison are highlighted in **bold**.

| #Bits | Method | DeepSeekV2-Lite | | | Qwen1.5-MoE-A2.7B | | | Qwen3-30B-A3B | | | Mixtral-8×7B | | |
|---|---|---|---|---|---|---|---|---|---|---|---|---|---|
| | | WT2$^\downarrow$ | C4$^\downarrow$ | 0-shot$_7^\uparrow$ | WT2$^\downarrow$ | C4$^\downarrow$ | 0-shot$_7^\uparrow$ | WT2$^\downarrow$ | C4$^\downarrow$ | 0-shot$_7^\uparrow$ | WT2$^\downarrow$ | C4$^\downarrow$ | 0-shot$_7^\uparrow$ |
| 16.0 | FP | 6.31 | 9.32 | 64.24 | 7.22 | 10.01 | 62.32 | 8.71 | 14.06 | 71.57 | 3.84 | 7.40 | 70.97 |
| 2.5 | Uniform | 8.14 | 13.59 | 59.81 | 10.36 | 20.67 | 44.12 | 10.29 | 17.45 | 65.58 | 6.10 | 10.35 | **65.49** |
| | PMQ | 6.95 | 10.74 | 59.60 | 9.01 | 13.57 | 54.57 | 10.94 | 17.01 | 61.96 | 5.10 | 9.21 | 64.34 |
| | GEMQ | **6.65** | **10.51** | **60.73** | **8.02** | **12.25** | **58.38** | **9.39** | **15.19** | **65.69** | **5.03** | **9.02** | 65.13 |
| 2.0 | Uniform | 9.57 | 16.90 | 50.06 | 14.25 | 34.59 | 41.06 | 10.97 | 19.05 | **61.28** | 6.29 | 11.73 | 55.72 |
| | PMQ | 7.99 | 12.55 | 52.52 | 10.47 | 16.29 | 52.41 | 13.91 | 19.84 | 51.39 | 6.10 | 11.36 | **60.38** |
| | GEMQ | **7.27** | **11.93** | **53.80** | **8.79** | **14.44** | **54.87** | **10.34** | **16.77** | 60.53 | **6.03** | **10.89** | 59.98 |
| 1.5 | Uniform | 15.37 | 32.04 | 45.07 | 50.73 | 169.65 | 35.96 | 21.09 | 47.80 | 49.42 | 10.67 | 25.39 | 47.45 |
| | PMQ | 11.05 | 18.31 | 46.91 | 14.10 | 23.88 | 47.58 | 20.85 | 34.59 | 42.08 | 8.47 | 20.77 | 51.78 |
| | GEMQ | **9.15** | **16.64** | **47.27** | **12.69** | **21.63** | **50.55** | **12.16** | **20.46** | **52.45** | **7.93** | **16.20** | **52.00** |

*Table 2.* Comparison with state-of-the-art methods on Mixtral-8×7B. 0-shot denotes the average accuracy on the specific task sets used by the respective baselines. Subscripts denote the performance gap relative to the FP model. W-A: weight-activation.

| Method | #Bits (W-A) | WT2$^\downarrow$ | C4$^\downarrow$ | 0-shot$^\uparrow$ |
|---|---|---|---|---|
| SpQR | 2.5-16 | 5.40 | 9.35 | 64.92$_{(-8.52\%)}$ |
| GEMQ | 2.5-16 | **5.03** | **9.02** | **65.13**$_{(-8.23\%)}$ |
| SpQR | 1.5-16 | Inf | Inf | 31.87$_{(-55.09\%)}$ |
| GEMQ | 1.5-16 | 7.93 | 16.20 | 52.00$_{(-26.73\%)}$ |
| MoEQuant | 3-16 | 4.90 | 8.24 | 57.24$_{(-13.29\%)}$ |
| GEMQ | 3-16 | **4.37** | **8.06** | **59.49**$_{(-6.90\%)}$ |
| EAQuant | 3-4 | 5.27 | 8.23 | 71.23$_{(-7.0\%)}$ |
| GEMQ | 3-16 | **4.37** | **8.06** | **76.10**$_{(-2.0\%)}$ |
| GEMQ | 2.5-16 | 5.03 | 9.02 | 72.52$_{(-6.6\%)}$ |

*Table 3.* Comparison of Mixtral-8×7B quantization on different calibration datasets. "M" denotes the MATH dataset.

| Method | Calib. | WT2$^\downarrow$ | C4$^\downarrow$ | 0-shot$_7^\uparrow$ | GSM8K$^\uparrow$ |
|---|---|---|---|---|---|
| FP | – | 3.84 | 7.40 | 70.97 | 57.77 |
| | | | 2.5 bits | | |
| PMQ | C4 | 5.10 | 9.21 | 64.34 | 33.06 |
| GEMQ | C4 | **5.03** | **9.02** | 65.13 | 31.77 |
| PMQ | M+C4 | 5.20 | 9.25 | 65.23 | 41.93 |
| GEMQ | M+C4 | 5.18 | 9.17 | **66.47** | **42.30** |
| | | | 2.0 bits | | |
| PMQ | C4 | 6.10 | 11.36 | 60.38 | 19.48 |
| GEMQ | C4 | 6.03 | 10.89 | 59.98 | 12.89 |
| PMQ | M+C4 | 6.16 | 11.25 | 60.20 | **23.84** |
| GEMQ | M+C4 | **6.03** | **10.77** | **61.07** | 23.12 |

In our experiments, we use *bits per expert* (bpe) to denote the expert quantization budget, defined as bpe = (total bits assigned) / (number of experts in the model). Following prior work, we consider four settings spanning high to low bit-widths: 3.0, 2.5, 2.0, and 1.5 bpe. We employ group-wise asymmetric GPTQ quantization (group size 128), using 128 random sequences of length 2048 from the WikiText2 training set for calibration. For router fine-tuning, we use the same calibration set as in quantization. We use the AdamW optimizer (Loshchilov & Hutter, 2017) to mini-mize cross-entropy loss with learning rate 1e−4, batch size 1, and weight decay 1e−4, while keeping all other settings identical to those used in pre-training. We perform one epoch of fine-tuning, since we observe that training gener-ally converges within a single epoch. All experiments are conducted on three NVIDIA H100-80GB GPUs.

### 5.1. Comparison of MoE-LLM Quantization Methods

**Baselines.** We evaluate against a uniform quantization baseline that assigns the same bit-width to all experts (Li et al., 2024; Chen et al., 2025b). For the 2.5/1.5-bit set-

tings, experts in the first half of the layers are quantized to 3/2 bits, while those in the second half are quantized to 2/1 bits. We further compare against state-of-the-art mixed-precision methods, including SpQR (Dettmers et al., 2023) for dense LLMs and PMQ (Huang et al., 2024a) for MoE-LLMs. Note that SpQR targets sub-tensor-level mixed precision within each weight tensor rather than expert-level quantization. We also compare against two state-of-the-art uniform MoE-LLM quantization methods, MoEQuant (Hu et al., 2025) and EAQuant (Fu et al., 2025), which employ advanced techniques such as calibration optimization and outlier suppression to improve quantization performance.

As shown in Tab. 1, mixed-precision approaches generally outperform the naive uniform quantization baseline, par-ticularly in low-bit settings, highlighting the importance of accounting for heterogeneous expert importance. Com-pared with the layer-wise local allocation method PMQ, the proposed GEMQ leverages global expert importance and mitigates router distortion, leading to better performance. As shown in Tab. 2, SpQR collapses under extreme low-bit settings due to aggressive quantization across all experts,

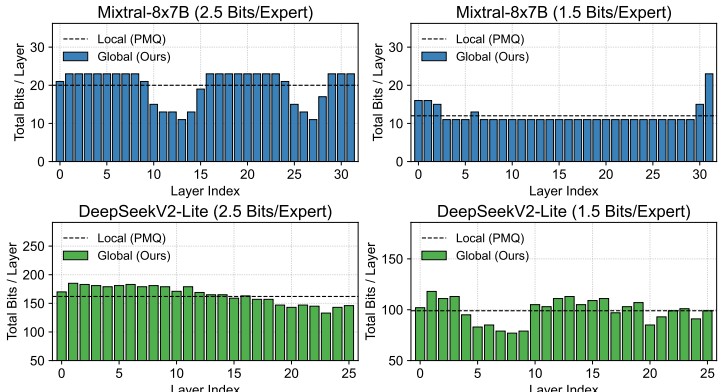

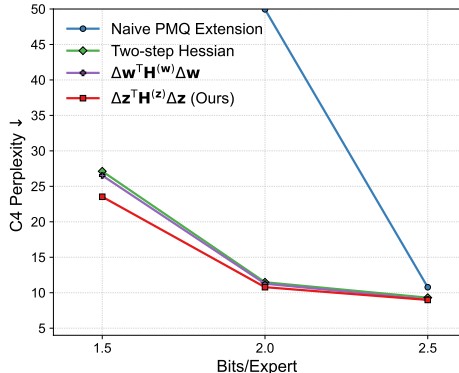

*(a)* Distribution of assigned bit-widths. Our method takes advantage of the layer-wise variation in expert importance.

*(b)* Comparison of global expert bit-width allocation methods on Mixtral-8×7B.

*Figure 4.* Analysis of global expert bit-width allocation.

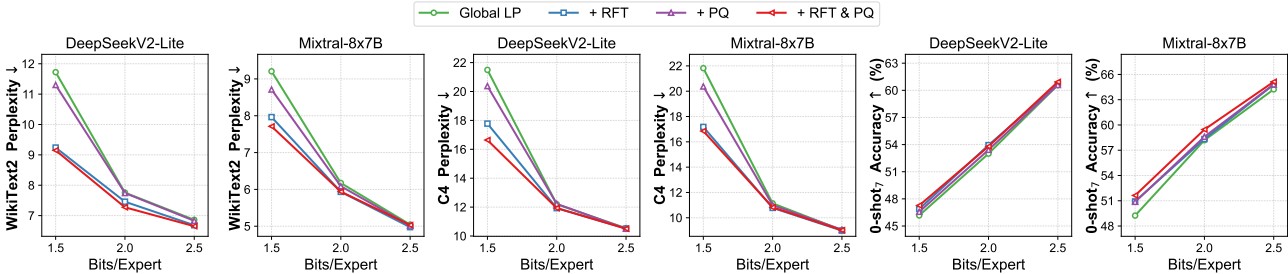

*Figure 5.* Ablation of the proposed techniques. "RFT" denotes global router fine-tuning, and "PQ" denotes progressive quantization. "Zero-shot Accuracy" is averaged over seven tasks.

whereas GEMQ preserves sufficient precision for critical experts, preventing collapse and maintaining performance. Additionally, GEMQ consistently outperforms both EAQuant and MoEQuant at comparable bit budgets, demonstrating the effectiveness of mixed-precision quantization. We provide full results and additional discussion in Appendix B.

In addition to general text modeling and commonsense tasks, we also evaluate performance on the challenging math-reasoning benchmark GSM8K (Cobbe et al., 2021). As shown in Tab. 3, when calibrating on the generic corpus C4 alone, both PMQ and GEMQ exhibit a noticeable accuracy drop, indicating a domain bias between the calibration data and the test data. This effect is more pronounced for our GEMQ because it does not heavily rely on heuristic regularization (*e.g.*, the $\alpha, \beta, \gamma$ hyperparameters on LP coefficients or the fixed per-layer bit budget). Nevertheless, this issue can be effectively mitigated by using more balanced and representative calibration data. Specifically, we add another 128 sequences from the MATH dataset (Hendrycks et al., 2021) and combine them with the original 128 C4 sequences for calibration and fine-tuning. As shown in Tab. 3, GEMQ substantially benefits from this simple mixed-data calibration setup: performance on the challenging GSM8K math-reasoning benchmark improves significantly. More importantly, under this setting, GEMQ outperforms the local-

based PMQ method on text modeling and commonsense tasks, while matching its performance on GSM8K. This highlights the stronger optimization capability brought by global bit allocation.

## 5.2. Quantization Overhead Analysis

In this section, we analyze the offline quantization overhead introduced by GEMQ and compare it with the statistic-based local method PMQ. Tab. 4 reports the wall-clock time and peak GPU memory usage of each component in a single quantization stage. As shown, the core components of GEMQ, including gradient computation and router fine-tuning, introduce only a minor runtime overhead of less than 3 minutes per stage (roughly 6% of the total runtime), while necessitating a relatively large increase in GPU memory usage due to gradient computation.

For progressive quantization to lower bit-widths, this overhead accumulates across stages but remains modest: three stages $(2.5 \rightarrow 2.0 \rightarrow 1.5)$ are sufficient to achieve strong 1.5-bit performance, with a total runtime of around two hours. Importantly, this is a *one-time offline cost* and introduces no additional inference-time latency, making it negligible relative to the sustained efficiency gains after quantization.

The proposed multi-stage progressive quantization further

*Table 4.* Resource breakdown for single-stage 2-bit quantization of Mixtral-8×7B and DeepSeek-V2-Lite. Wall-clock time (seconds) and peak GPU memory usage (GB) measured on NVIDIA H100-80GB GPUs.

| Model | Resource | Method | Comp. Grad. | Comp. Stats | Solving LP | Quant. (GPTQ) | FT Routers | Total |
|-------|----------|--------|-------------|-------------|------------|---------------|------------|-------|
| **Mixtral-8×7B** | Time | PMQ | N/A | $455.6_{(21.1\%)}$ | $0.1_{(0.0\%)}$ | $1706.3_{(78.9\%)}$ | N/A | $2162.0_{(100\%)}$ |
| | | GEMQ | $84.3_{(3.6\%)}$ | $468.2_{(20.3\%)}$ | $0.1_{(0.0\%)}$ | $1698.2_{(73.5\%)}$ | $59.4_{(2.6\%)}$ | $2310.2_{(100\%)}$ |
| | Mem. | PMQ | N/A | 31.8 | N/A | 21.5 | N/A | $31.8_{(max)}$ |
| | | GEMQ | 191.2 | 35.1 | N/A | 21.5 | 136.3 | $191.2_{(max)}$ |
| **DeepSeek-V2-Lite** | Time | PMQ | N/A | $721.3_{(27.3\%)}$ | $0.2_{(0.0\%)}$ | $1923.7_{(72.7\%)}$ | N/A | $2645.2_{(100\%)}$ |
| | | GEMQ | $130.5_{(4.6\%)}$ | $725.3_{(25.4\%)}$ | $0.2_{(0.0\%)}$ | $1920.2_{(67.3\%)}$ | $76.7_{(2.7\%)}$ | $2852.9_{(100\%)}$ |
| | Mem. | PMQ | N/A | 30.5 | N/A | 8.8 | N/A | $30.5_{(max)}$ |
| | | GEMQ | 82.1 | 32.8 | N/A | 8.8 | 55.2 | $82.1_{(max)}$ |

*Table 5.* Comparison of perplexity$^{\downarrow}$ for different router calibration methods on Mixtral-8×7B 1.5-bit quantization.

| Method | WikiText2 | C4 |
|--------|-----------|-----|
| FP | 3.84 | 7.40 |
| w/o Calibration | 9.66 | 23.53 |
| w/ FP Router Logits | 9.29 | 22.72 |
| w/ Router Fine-Tuned | **7.69** | **17.18** |

*Table 6.* Perplexity$^{\downarrow}$ and average router change ratio of 1.5-bit quantized Mixtral-8×7B, using different expert importance estimation models.

| Estimation Model | WikiText2 | C4 | Change Ratio |
|------------------|-----------|-----|--------------|
| FP | 9.66 | 23.53 | 41.31% |
| 4-bit | 9.39 | 22.92 | 39.10% |
| 2-bit | 9.33 | 22.80 | 34.69% |
| 2-bit (RFT) | **9.01** | **21.88** | 38.38% |

offers a flexible accuracy–overhead trade-off: when the runtime budget is limited, the number of stages can be further reduced, or gradients and statistics can be computed directly from the full-precision model in a single stage, at the cost of modest performance degradation.

### 5.3. Ablation Study

**Global Expert Bit-width Allocation.** In the previous section, we demonstrated the advantage of global expert allocation compared with its local counterpart. Here, we provide further analysis by visualizing the layer-wise distribution of assigned bit-widths in Fig. 4a. As shown, our method effectively leverages variations in layer importance, allocating more bits to critical layers and thereby achieving better performance. Moreover, the results reveal that layer importance varies across models and bit budgets, underscoring the need for our automatic LP-based formulation to achieve flexible and effective bit allocation. In Fig. 4b, we compare the proposed method with several alternative global expert bit-width allocation strategies. A naive extension of PMQ that applies layer-wise coefficients in a global LP suffers significant degradation, as the resulting expert importance is not comparable across layers. A two-step scheme that first assigns layer-level bit budgets and then allocates bits to experts yields better results but still underperforms global allocation methods. Finally, we evaluate a variant that applies the FIM approximation directly to the weight Hessian. While this approach performs well in high-bit settings, it degrades significantly under low-bit quantization, suggesting that the output transformation in Eq. 4 provides a more reliable proxy for expert importance estimation.

We further evaluate robustness to calibration subset sampling and size. Fig. 6 presents expert importance and layer-wise bit allocation across three randomly sampled C4 subsets with 128 sequences and one with 2048 sequences, while Tab. 15 reports quantization performance. Tab. 16 shows performance saturation at approximately 128 sequences. Overall, these results demonstrate that GEMQ is robust to sampling noise and subset size, preserving consistent expert importance and stable performance. Appendix D presents additional ablations on extra LP constraints and bit-width candidate choices.

**Global Router Fine-Tuning.** Fig. 5 compares GEMQ with and without router fine-tuning after quantization. As shown, global router fine-tuning substantially reduces perplexity on both the WikiText2 and C4, especially in the challenging 1.5 bpe setting where router logits are heavily distorted, and also improves zero-shot accuracy on downstream tasks. Tab. 5 compares our global router fine-tuning strategy with layer-wise rigid calibration (Fu et al., 2025; Chen et al., 2025b), which enforces routers to mimic the FP distribution. As seen, even when router logits in quantized models are replaced with recorded FP logits during inference, performance gains remain marginal, indicating that FP-matching is suboptimal after expert weight changes. In contrast, our task-loss-based global router fine-tuning yields substantial improvements. We further ablate fine-tuning settings (*e.g.*, training epochs and calibration sample size) and analyze their effects in Appendix E.

**Progressive Quantization.** Fig. 5 and Tab. 21 show that PQ improves both perplexity and zero-shot accuracy across models, with further gains when combined with router

*Table 7.* Decoding performance (batch=1) of GEMQ-quantized models on a single NVIDIA H100-80G GPU. Maximum memory consumption is reported. Speed is measured in tokens per second.

| #Bits | DeepSeekV2-Lite | | Mixtral-8×7B | |
|---|---|---|---|---|
| | Mem (GB) | Speed (tok/s) | Mem (GB) | Speed (tok/s) |
| 16.0 | 31.6 | 159.4 | OOM | - |
| 2.5 | $7.2_{(23\%)}$ | $229.8_{(1.4\times)}$ | 17.8 | 82.5 |
| 2.0 | $6.2_{(20\%)}$ | $237.1_{(1.5\times)}$ | 14.9 | 84.6 |
| 1.5 | $5.2_{(16\%)}$ | $248.4_{(1.6\times)}$ | 12.0 | 88.6 |

fine-tuning. As analyzed in Sec. 4.3, using a quantized model closer to the target bit budget reduces expert selection changes and mitigates abrupt loss shifts. Consistent with this, Tab. 6 and Fig. 9 show that, for 1.5-bit quantization, using a 2-bit model for importance estimation reduces the average expert selection change rate from 41.31% to 34.69% compared to the FP model, leading to more accurate importance estimation and lower perplexity. Fine-tuning the routers of the 2-bit quantized model and using it for importance estimation further improves performance, consistent with our analysis that better satisfaction of the zero-gradient assumption yields a more accurate approximation.

## 6. Memory Saving and Inference Efficiency

We leverage HQQ (Badri & Shaji, 2023) for bit-packing and storage of quantized weights. For inference acceleration, we implement a fused Triton (Tillet et al., 2019) kernel for MoE blocks that supports fast low-bit (*i.e.*, 1-, 2-, and 3-bit) dequantization and grouped GEMM, optimized for the memory-bound decoding stage. As shown in Tab. 7, compared with a strong FP16 baseline with `torch.compile` enabled, the quantized DeepSeekV2-Lite model achieves up to 6.1× memory reduction and 1.6× decoding speedup on an NVIDIA H100 GPU. Notably, the full-precision Mixtral-8×7B model exceeds the memory capacity of a single GPU (over 90 GB), whereas the 2.5-bit quantized version reduces peak runtime memory to 18 GB, enabling deployment even on consumer-grade hardware. More detailed comparisons of model size reduction and inference speed across quantized models are reported in Appendix C.

## 7. Limitations

While GEMQ achieves strong performance across diverse MoE-LLM families, two main limitations remain. First, the effectiveness of GEMQ depends on the choice of calibration data. Although the method is generally robust to common calibration sets, performance on specialized downstream tasks (*e.g.*, mathematical reasoning) can benefit from domain-aligned samples. A more principled study of calibration data construction is left for future work. Second, the gradient computation required for expert importance esti-

mation and router fine-tuning introduces a non-trivial peak GPU memory footprint that scales with model size and may exceed single-GPU capacity for very large MoE models, requiring multi-GPU setups or memory-saving techniques during the offline quantization stage.

## 8. Conclusion

In this work, we identify two key limitations in existing mixed-precision quantization methods for MoE-LLMs and propose GEMQ, which enables principled expert bit-width allocation before quantization and effective router adaptation after quantization. We further introduce a progressive quantization framework that integrates both techniques to improve expert importance estimation in low-bit scenarios. GEMQ stands as a pioneering framework that validates the potential of global expert importance for mixed-precision MoE quantization. By formulating the allocation problem globally, GEMQ unlocks a substantially richer optimization space than local methods. Our empirical results provide compelling evidence that global allocation is a promising and effective approach for optimizing the compression-performance trade-off in MoE-LLMs.

## Acknowledgements

This work was supported in part by the National Science Foundation under Grants CNS-2328972, CCF-2324937, CNS-2122320, and CNS-2133267. This work was also supported in part by the TetraMem Inc. Research Award. We acknowledge the computational resources provided by the Pittsburgh Supercomputing Center and the National Center for Supercomputing Applications.

## Impact Statement

This paper presents advances in machine learning through a method that improves the memory efficiency and deployability of large mixture-of-experts models via global mixed-precision quantization. By enabling lightweight and resource-efficient inference while maintaining reasonable accuracy, this technique has the potential to broaden access to large-scale language models across a wider range of computational environments and applications. We do not foresee direct negative social consequences that require specific discussion. Overall, our work aims to foster practical adoption and responsible scaling of machine learning systems by reducing computational barriers for researchers, developers, and practitioners.

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

# A. Details on Models and Evaluation

*Table 8.* Details of MoE-LLMs used in evaluation. "Expert Prop." and "Router Prop." denote the percentage of experts and routers in the total number of parameters ("#Params"), respectively. For the "#Experts" column, we follow the convention (#Routed Experts + #Shared Experts) × #Layers. Qwen3-30B-A3B and Mixtral-8×7B contain only routed experts and no shared experts. For DeepSeekV2-Lite, the first dense layer is omitted in the #Layers count. Top-K experts are selected from the routed experts.

| Model | #Total Params. | #Active Params. | Expert Prop. | Router Prop. | #Experts | Top-K |
|---|---|---|---|---|---|---|
| DeepSeekV2-Lite | 15.7B | 2.4B | 94.51% | 0.021% | (64+2)×26 | 6 |
| Qwen1.5-MoE | 14.3B | 2.7B | 92.82% | 0.021% | (60+4)×24 | 4 |
| Qwen3-30B-A3B | 30.5B | 3.3B | 94.95% | 0.041% | 128×48 | 8 |
| Mixtral-8×7B | 46.7B | 12.9B | 97.00% | 0.002% | 8×32 | 2 |

To comprehensively evaluate the proposed GEMQ for MoE-LLMs quantization, we select four representative MoE-LLMs with varying sizes, architectures, and characteristics: DeepseekV2-Lite (Dai et al., 2024), Qwen1.5-MoE-A2.7B (Yang et al., 2025), Qwen3-30B-A3B (Yang et al., 2025), and Mixtral-8×7B (Jiang et al., 2024). The details of these models are summarized in Tab. 8. Note that, except for Qwen3-30B-A3B, all models are only pre-trained for language modeling without supervised fine-tuning (SFT). Qwen3-30B-A3B undergoes both pre-training and post-training.

In addition to evaluating perplexity on general language modeling benchmarks, we evaluate different quantization methods on seven zero-shot tasks: PIQA (Bisk et al., 2020), ARC-Easy (Clark et al., 2018), ARC-Challenge (Clark et al., 2018), HellaSwag (Zellers et al., 2019), WinoGrande (Sakaguchi et al., 2021), MathQA (Amini et al., 2019), and MMLU (Hendrycks et al., 2020). We also perform 5-shot evaluation on MMLU to assess the in-context learning ability of the models. Moreover, we evaluate our method on the challenging GSM8K (Cobbe et al., 2021) benchmark to assess the mathematical reasoning ability of quantized models. All benchmark results are obtained using LM-Evaluation-Harness (v0.4.8) (Gao et al., 2024). We report `acc_norm` when available; otherwise, `acc` is reported.

# B. Comparison with State-of-the-Art Methods

In this section, we present the full results from Tab. 1 and Tab. 2, along with additional discussion comparing state-of-the-art MoE-LLM quantization methods.

**Comparison with Mixed-Precision Quantization Methods.** Tab. 9 reports full evaluation results on general text modeling and zero-shot downstream tasks, compared against a naive uniform quantization baseline and the mixed-precision MoE-LLM method PMQ (Huang et al., 2024a). Tab. 10 further presents performance on the 5-shot MMLU benchmark. For a fair comparison, we vary only the bit-widths of experts while keeping all attention modules fixed at 4 bits, and apply GPTQ with identical quantization settings across all methods. Across both tables, GEMQ consistently outperforms the uniform baseline and PMQ in most cases, demonstrating strong preservation of knowledge, reasoning ability, and few-shot in-context learning performance across diverse academic and professional domains.

Additionally, we compare against SpQR (Dettmers et al., 2023), a mixed-precision quantization method originally designed for dense LLMs. SpQR applies fine-grained sub-tensor-level mixed precision by retaining a small subset of outlier weights in full precision (16-bit) while quantizing the remaining weights to low precision. We evaluate GEMQ and SpQR on Mixtral-8×7B using the same group size (128) and calibration data (128 sequences from WikiText-2). All attention modules are quantized to 4-bit, while expert modules are allocated bits according to the target budget. As shown in Tab. 11, our method slightly outperforms SpQR in the higher-bit regime. Under extreme low-bit settings, although SpQR retains a small set of 16-bit parameters in each expert, the vast majority of parameters are quantized to 1-bit, causing the model to collapse. In contrast, our expert-level mixed-precision method maintains sufficient precision for critical experts and, combined with router fine-tuning, prevents collapse and preserves performance.

**Comparison with Other MoE-LLM Quantization Methods.** We compare GEMQ with two state-of-the-art uniform MoE-LLM quantization methods, EAQuant (Fu et al., 2025) and MoEQuant (Hu et al., 2025). EAQuant primarily focuses on outlier suppression under uniform weight-activation quantization, whereas GEMQ derives a global mixed-precision strategy for weight-only quantization. Although EAQuant also considers router distribution shift, it adopts a layer-wise rigid alignment scheme that yields only marginal gains (*e.g.*, < 0.1 perplexity improvement reported in their paper). In contrast, GEMQ employs global router fine-tuning and achieves substantially larger improvements, as shown in Fig. 5, Tab. 5, and

*Table 9.* Comparison of perplexity$^\downarrow$ on the WikiText2 (WT2) and C4 test sets, and accuracy$^\uparrow$ (%) across seven zero-shot tasks: PIQA (PQ), ARC-Easy (AE), ARC-Challenge (AC), HellaSwag (HS), WinoGrande (WG), MathQA (MQ), and MMLU (MM). The average accuracy (Avg.) over zero-shot tasks is also reported. #Bits denotes the average bit-width per expert. Model parameters are reported as (#Total - A#Activated). The best results in each comparison are highlighted in **bold**.

| Model | #Bits | Method | WT2$^\downarrow$ | C4$^\downarrow$ | PQ$^\uparrow$ | AE$^\uparrow$ | AC$^\uparrow$ | HS$^\uparrow$ | WG$^\uparrow$ | MQ$^\uparrow$ | MM$^\uparrow$ | Avg.$^\uparrow$ |
|---|---|---|---|---|---|---|---|---|---|---|---|---|
| DeepSeekV2-Lite (15.7B - A2.4B) | 16.0 | FP | 6.31 | 9.32 | 80.09 | 76.47 | 48.98 | 78.01 | 71.51 | 39.73 | 54.90 | 64.24 |
| | 2.5 | Uniform | 8.14 | 13.59 | 76.66 | 71.34 | 45.22 | 72.44 | 69.46 | 34.77 | **48.78** | 59.81 |
| | | PMQ | 6.95 | 10.74 | 77.86 | 71.68 | 44.62 | 72.76 | 69.06 | **36.18** | 45.01 | 59.60 |
| | | GEMQ | **6.65** | **10.51** | **77.91** | **74.41** | **46.93** | **73.39** | 69.46 | 35.61 | 47.37 | **60.73** |
| | 2.0 | Uniform | 9.57 | 16.90 | 73.07 | 60.61 | 35.84 | 62.87 | 62.04 | 28.74 | 27.25 | 50.06 |
| | | PMQ | 7.99 | 12.55 | 75.14 | 61.20 | 37.37 | 66.20 | **68.11** | 30.55 | **29.06** | 52.52 |
| | | GEMQ | **7.27** | **11.93** | **76.55** | **68.10** | **40.53** | **68.43** | 66.38 | 27.97 | 28.61 | **53.80** |
| | 1.5 | Uniform | 15.37 | 32.04 | 67.08 | 53.83 | 29.78 | 51.94 | 60.38 | **25.73** | **26.73** | 45.07 |
| | | PMQ | 11.05 | 18.31 | **71.65** | 53.58 | 30.63 | **57.19** | **64.64** | 24.12 | 26.56 | 46.91 |
| | | GEMQ | **9.15** | **16.64** | 71.55 | **58.50** | **32.08** | 56.40 | 62.90 | 25.39 | 24.05 | **47.27** |
| Qwen1.5-MoE (14.3B - A2.7B) | 16.0 | FP | 7.22 | 10.01 | 80.52 | 69.23 | 44.11 | 77.21 | 68.59 | 35.24 | 61.31 | 62.32 |
| | 2.5 | Uniform | 10.36 | 20.67 | 71.49 | 49.07 | 29.44 | 53.98 | 55.96 | 23.65 | 25.25 | 44.12 |
| | | PMQ | 9.01 | 13.57 | 77.58 | 60.56 | 36.26 | 70.69 | 65.82 | 28.68 | 42.38 | 54.57 |
| | | GEMQ | **8.02** | **12.25** | **78.40** | **66.96** | **42.92** | **71.78** | 65.82 | **30.62** | **52.14** | **58.38** |
| | 2.0 | Uniform | 14.25 | 34.59 | 66.38 | 42.89 | 30.29 | 49.06 | 53.12 | 21.78 | 23.92 | 41.06 |
| | | PMQ | 10.47 | 16.29 | 74.76 | 59.05 | 34.56 | 64.34 | 64.72 | **28.41** | 41.01 | 52.41 |
| | | GEMQ | **8.79** | **14.44** | **76.39** | **61.36** | **36.69** | **67.00** | **68.11** | 27.20 | **47.37** | **54.87** |
| | 1.5 | Uniform | 50.73 | 169.65 | 58.32 | 33.8 | 25.09 | 38.19 | 51.38 | 21.81 | 23.16 | 35.96 |
| | | PMQ | 14.10 | 23.88 | 70.46 | 55.77 | 31.91 | 54.55 | 62.51 | **24.99** | 32.88 | 47.58 |
| | | GEMQ | **12.69** | **21.63** | **74.81** | **58.71** | **35.41** | **58.65** | **63.38** | 23.38 | **39.50** | **50.55** |
| Qwen3-30B-A3B (30.5B - A3.3B) | 16.0 | FP | 8.71 | 14.06 | 80.25 | 79.29 | 55.80 | 77.75 | 71.11 | 58.89 | 77.88 | 71.57 |
| | 2.5 | Uniform | 10.29 | 17.45 | 78.07 | **75.93** | **50.07** | 71.72 | 68.01 | 42.52 | **72.72** | 65.58 |
| | | PMQ | 10.94 | 17.01 | 76.61 | 68.43 | 45.99 | 72.13 | 66.22 | 36.92 | 67.39 | 61.96 |
| | | GEMQ | **9.39** | **15.19** | **78.35** | 74.16 | 49.49 | **75.00** | **68.59** | **43.18** | 71.04 | **65.69** |
| | 2.0 | Uniform | 10.97 | 19.05 | 76.93 | **68.27** | **44.78** | 69.36 | **68.75** | **35.33** | **65.53** | **61.28** |
| | | PMQ | 13.91 | 19.84 | 73.18 | 53.96 | 35.84 | 70.05 | 64.17 | 24.22 | 38.29 | 51.39 |
| | | GEMQ | **10.34** | **16.77** | **76.93** | 65.11 | 43.77 | **72.22** | 68.11 | 34.34 | 63.21 | 60.53 |
| | 1.5 | Uniform | 21.09 | 47.80 | 69.21 | 57.74 | **37.88** | 53.44 | 62.98 | **25.43** | 39.25 | 49.42 |
| | | PMQ | 20.85 | 34.59 | 58.32 | 34.64 | 24.23 | **62.72** | **65.98** | 21.91 | 26.76 | 42.08 |
| | | GEMQ | **12.16** | **20.46** | **74.37** | **58.59** | 35.07 | 61.20 | 63.14 | 25.06 | **49.69** | **52.45** |
| Mixtral-8×7B (46.7B - A12.9B) | 16.0 | FP | 3.84 | 7.40 | 83.68 | 83.42 | 59.73 | 83.99 | 76.32 | 41.78 | 67.87 | 70.97 |
| | 2.5 | Uniform | 6.10 | 10.35 | 80.03 | 75.93 | **52.73** | 79.09 | 73.56 | 34.84 | **62.26** | **65.49** |
| | | PMQ | 5.10 | 9.21 | 80.36 | 74.87 | 51.28 | 79.18 | **73.95** | **34.94** | 55.78 | 64.34 |
| | | GEMQ | **5.03** | **9.02** | **81.07** | **75.93** | 51.79 | **79.57** | 73.72 | 34.10 | 59.74 | 65.13 |
| | 2.0 | Uniform | 6.29 | 11.73 | 75.14 | 64.73 | 43.77 | 71.31 | 67.72 | 29.15 | 38.23 | 55.72 |
| | | PMQ | 6.10 | 11.36 | **78.29** | **73.15** | **49.66** | **73.55** | **71.35** | **30.65** | 46.03 | **60.38** |
| | | GEMQ | **6.03** | **10.89** | 77.53 | 71.42 | 46.50 | 72.40 | 70.40 | 30.39 | **51.24** | 59.98 |
| | 1.5 | Uniform | 10.67 | 25.39 | 65.61 | 56.10 | 34.13 | 55.77 | 61.88 | 24.99 | 33.66 | 47.45 |
| | | PMQ | 8.47 | 20.77 | 73.01 | 63.13 | **37.37** | **64.16** | 65.98 | **27.04** | 31.76 | 51.78 |
| | | GEMQ | **7.93** | **16.20** | **73.67** | **63.47** | 36.95 | 59.93 | **67.25** | 26.20 | **36.50** | **52.00** |

*Table 10.* Comparison of 5-shot MMLU accuracy$^\uparrow$ (%). #Bits indicates the average bits per expert.

| #Bits | Method | DeepSeekV2-Lite | Mixtral-8×7B |
|---|---|---|---|
| 16.0 | FP | 58.16 | 70.37 |
| 2.5 | Uniform | **54.09** | 50.51 |
| | PMQ | 49.98 | 60.70 |
| | GEMQ | 53.58 | **63.21** |
| 2.0 | Uniform | 27.67 | 51.17 |
| | PMQ | **38.60** | 46.79 |
| | GEMQ | 35.00 | **54.55** |
| 1.5 | Uniform | 28.97 | 34.31 |
| | PMQ | 29.12 | 34.53 |
| | GEMQ | **29.70** | **36.76** |

*Table 11.* Comparison with SpQR and GEMQ on Mixtral-8×7B. We show the perplexity results >1000 by Inf.

| #Bits | Method | WikiText2$^\downarrow$ | C4$^\downarrow$ | 0-shot$_7^\uparrow$ |
|---|---|---|---|---|
| 16.0 | FP | 3.84 | 7.40 | 70.97 |
| 2.5 | SpQR | 5.40 | 9.35 | 64.92 |
| | GEMQ | **5.03** | **9.02** | **65.13** |
| 1.5 | SpQR | Inf | Inf | 31.87 |
| | GEMQ | **7.93** | **16.20** | **52.00** |

*Table 12.* Comparison between EAQuant and GEMQ on Mixtral-8×7B quantization. Subscripts denote the performance gap (in %) relative to the FP baseline. "*" denotes results from the original paper. #Bits denotes the bit-width used for weight–activation quantization.

| Method | #Bits | WikiText2$^\downarrow$ | C4$^\downarrow$ | PIQA$^\uparrow$ | ARC-E$^\uparrow$ | ARC-C$^\uparrow$ | BoolQ$^\uparrow$ | Winogrande$^\uparrow$ | 0-shot Avg.$^\uparrow$ |
|---|---|---|---|---|---|---|---|---|---|
| FP16* | 16-16 | 3.84 | 6.98 | 83.41 | 83.29 | 55.80 | 84.56 | 75.85 | 76.58 |
| EAQuant* | 3-4 | 5.27$_{(+37.2)}$ | 8.23$_{(+17.9)}$ | 79.05$_{(-5.2)}$ | 78.45$_{(-5.8)}$ | 50.68$_{(-9.2)}$ | 78.69$_{(-6.9)}$ | 69.30$_{(-8.6)}$ | 71.23$_{(-7.0)}$ |
| FP16 | 16-16 | 3.84 | 7.40 | 83.68 | 83.42 | 59.73 | 85.05 | 76.32 | 77.64 |
| GEMQ | 3-16 | **4.37**$_{(+13.8)}$ | **8.06**$_{(+9.0)}$ | **82.54**$_{(-1.4)}$ | **80.68**$_{(-3.3)}$ | **57.34**$_{(-4.0)}$ | **85.02**$_{(-0.1)}$ | **74.90**$_{(-1.9)}$ | **76.10**$_{(-2.0)}$ |
| GEMQ | 2.5-16 | 5.03$_{(+31.0)}$ | 9.02$_{(+21.9)}$ | 81.07$_{(-3.1)}$ | 75.93$_{(-9.0)}$ | 51.79$_{(-13.3)}$ | 80.09$_{(-5.8)}$ | 73.72$_{(-3.4)}$ | 72.52$_{(-6.6)}$ |

*Table 13.* Comparison between MoEQuant and GEMQ on Mixtral-8×7B quantization. Subscripts denote the performance gap (in %) relative to the FP baseline. "*" denotes results from the original paper. #Bits denotes the bit-width used for weight–activation quantization.

| Method | #Bits | WikiText2$^\downarrow$ | C4$^\downarrow$ | BoolQ$^\uparrow$ | MathQA$^\uparrow$ | MMLU$^\uparrow$ | GSM8K$^\uparrow$ | 0-shot Avg.$^\uparrow$ |
|---|---|---|---|---|---|---|---|---|
| FP16* | 16-16 | 3.84 | 6.87 | 85.23 | 42.41 | 70.50 | 65.88 | 66.01 |
| MoEQuant* | 3-16 | 4.90$_{(+27.6)}$ | 8.24$_{(+19.9)}$ | 82.81$_{(-2.8)}$ | 38.82$_{(-8.5)}$ | 64.10$_{(-9.1)}$ | 43.21$_{(-34.4)}$ | 57.24$_{(-13.3)}$ |
| FP16 | 16-16 | 3.84 | 7.40 | 85.05 | 41.78 | 70.97 | 57.77 | 63.89 |
| GEMQ | 3-16 | **4.37**$_{(+13.8)}$ | **8.06**$_{(+9.0)}$ | **85.02**$_{(-0.1)}$ | 38.63$_{(-7.5)}$ | 64.63$_{(-8.9)}$ | 49.66$_{(-14.0)}$ | **59.49**$_{(-6.9)}$ |
| GEMQ | 2.5-16 | 5.03$_{(+31.0)}$ | 9.02$_{(+21.9)}$ | 80.09$_{(-5.8)}$ | 34.10$_{(-18.4)}$ | 59.74$_{(-15.8)}$ | 42.30$_{(-26.8)}$ | 54.06$_{(-15.4)}$ |

Fig. 7. Moreover, EAQuant targets relatively high-bit regimes ($\geq$3 bpe), whereas GEMQ focuses on more aggressive low-bit settings ($\leq$2.5 bpe) to better address the memory footprint of expert parameters. MoEQuant constructs optimized calibration data for uniform weight-only quantization via self-sampling and extends GPTQ with affinity-guided weighting to reduce quantization error. However, it does not explore mixed-precision bit allocation or explicitly address router distortion induced by expert quantization, and similarly operates in higher-bit regimes ($\geq$3 bpe).

Tab. 12 and Tab. 13 present quantitative comparisons. Since EAQuant does not share identical quantization configurations with GEMQ, we compare our 2.5-bpe (W2.5A16) and 3.0-bpe (W3A16) models against EAQuant's closest available setting (W3A4). For MoEQuant, we compare against the W3A16 configuration. To account for minor discrepancies in FP16 baselines, we additionally report relative performance degradation as $\Delta = (\text{Quantized} - \text{FP16})/\text{FP16}$. The results show that GEMQ consistently outperforms both EAQuant and MoEQuant at comparable bit budgets, while achieving competitive performance even in more aggressive low-bit regimes.

# C. Memory Saving and Inference Efficiency

In this section, we report inference performance measured on a single NVIDIA H100-80G GPU. Since large LLMs, including MoE-LLMs, are predominantly memory-bound (Kim et al., 2023a; Lin et al., 2024), the primary bottleneck lies in the autoregressive decoding stage. Accordingly, we focus on optimizing the decoding stage, where low-bit weight-only quantization yields the most benefit by reducing memory traffic from HBM. As shown in Fig. 14, the quantized model substantially reduces peak memory usage during inference and achieves significant speedup. Specifically, quantizing Mixtral-8×7B to 2.5-bit reduces runtime memory to under 20 GB, enabling deployment on consumer-grade devices such as the NVIDIA RTX 4090 GPU.

Tab. 14 also reports results for models quantized with another mixed-precision method PMQ. In both setups, attention modules are quantized to 4-bit and router modules remain in 16-bit. We maintain the same average expert bit-width for both methods, ensuring that the only variable is the expert-wise bit allocation policy. Consequently, the observed differences in inference speed stem solely from expert selection for each token. As shown in Tab. 14, only minor differences in decoding memory usage and speed are observed between PMQ and GEMQ.

*Table 14.* Comparison of model decoding performance on a single NVIDIA H100-80G GPU. All inference experiments are conducted with batch size one. Max memory used in decoding is reported. Model sizes include quantization parameters (*i.e.*, scales and zero-points). Speed is measured for generating 200 new tokens (in tokens per second).

| #Bits | Method | DeepSeekV2-Lite | | | Mixtral-8×7B | | |
|---|---|---|---|---|---|---|---|
| | | Size(GB) | Mem(GB) | Speed (Tok/Sec) | Size(GB) | Mem(GB) | Speed(Tok/Sec) |
| 16.0 | FP | 29.91 | 31.6 | 159.4 | 87.0 | OOM | - |
| 2.5 | PMQ | 6.0 | 7.2 | 228.5 | 16.3 | 17.8 | 83.3 |
| | GEMQ | 6.0 | 7.2 | 229.8 | 16.3 | 17.8 | 82.5 |
| 2.0 | PMQ | 5.1 | 6.2 | 238.4 | 13.4 | 14.9 | 84.9 |
| | GEMQ | 5.1 | 6.2 | 237.1 | 13.4 | 14.9 | 84.6 |
| 1.5 | PMQ | 4.1 | 5.3 | 247.8 | 10.5 | 12.0 | 88.7 |
| | GEMQ | 4.1 | 5.2 | 248.4 | 10.5 | 12.0 | 88.6 |

# D. Further Analysis of Global Expert Bit-Width Allocation

*Table 15.* Average performance of GEMQ-quantized models on three randomly sampled calibration subsets (128 sequences of 2048 tokens each) from the C4 dataset.

| #Bits | DeepSeekV2-Lite | | | Mixtral-8×7B | | |
|---|---|---|---|---|---|---|
| | WikiText2$^{\downarrow}$ | C4$^{\downarrow}$ | 0-shot$^{\updownarrow}_{7}$ | WikiText2$^{\downarrow}$ | C4$^{\downarrow}$ | 0-shot$^{\updownarrow}_{7}$ |
| 2.5 | $6.65 \pm 0.01$ | $10.47 \pm 0.02$ | $60.72 \pm 0.21$ | $4.95 \pm 0.02$ | $8.92 \pm 0.03$ | $65.02 \pm 0.13$ |
| 2.0 | $7.26 \pm 0.07$ | $11.80 \pm 0.04$ | $54.43 \pm 0.27$ | $5.89 \pm 0.03$ | $10.70 \pm 0.04$ | $60.00 \pm 0.21$ |
| 1.5 | $9.22 \pm 0.21$ | $17.09 \pm 0.26$ | $47.08 \pm 0.35$ | $7.75 \pm 0.15$ | $16.04 \pm 0.21$ | $52.79 \pm 0.33$ |

**Robustness Test for Bit-Width Allocation.**   To examine how expert importance changes with different input subsets, we conduct robustness experiments using three random seeds to sample 128 sequences, as well as a larger subset of 2048 sequences, from the C4 dataset for bit allocation. We compare the per-expert-per-bit estimated errors (*i.e.*, Eq. 6, the proxy for expert importance) and the resulting layer-wise bit-allocation statistics in Fig. 6. The correspding averaged performance is reported in Tab. 15. As shown the figures, GEMQ is relatively robust to sampling noise, as the estimated error curves largely overlap even though only 128 sequences are used for calibration, achieving an average Pearson correlation over 0.99. Importantly, the key experts (*i.e.*, the peaks in the error-estimation curves) with large estimated errors are consistently identified across different samples. The overall expert-wise and layer-wise trends also remain closely aligned, indicating that expert importance is preserved across different calibration subsets. As a result, GEMQ yields consistent bit-allocation results and maintains model performance.

Tab. 16 shows the effect of calibration dataset size on quantization performance. As the dataset size increases, performance improves but quickly saturates around 128 sequences, indicating that 128 samples are sufficient to achieve stable calibration.

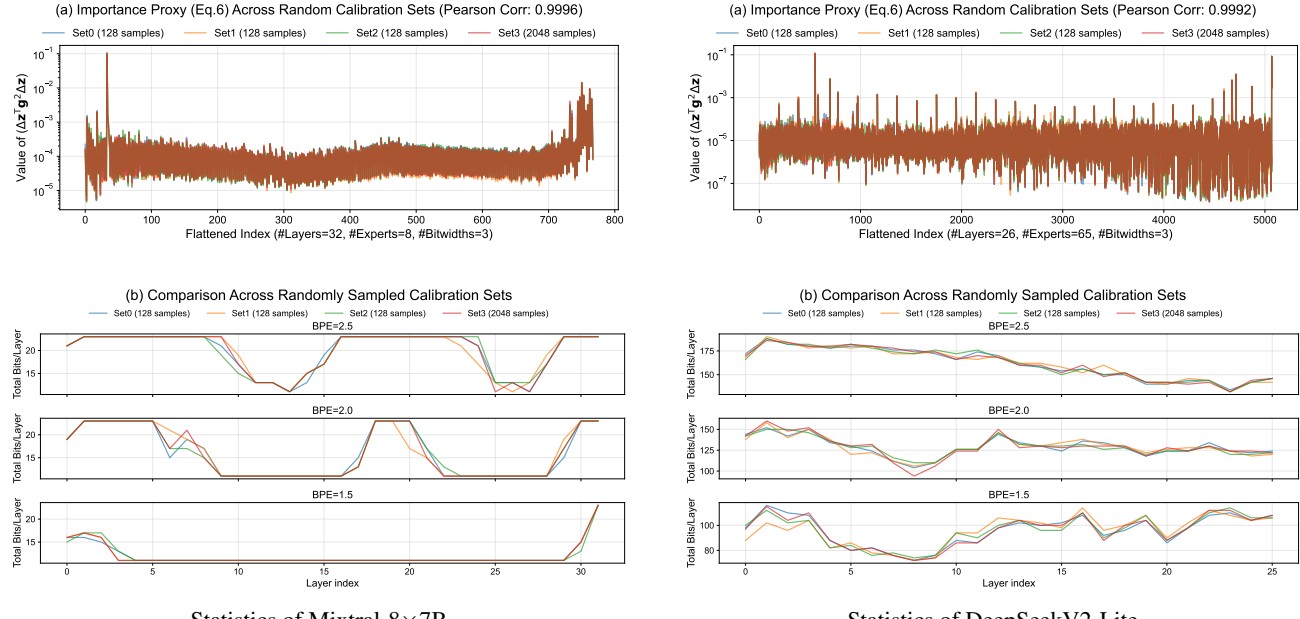

Statistics of Mixtral-8×7B        Statistics of DeepSeekV2-Lite

*Figure 6.* Statistics of the expert importance proxy and corresponding bit-width allocation across four randomly sampled calibration subsets from C4 (three with 128 sequences and one with 2048 sequences; each sequence contains 2048 tokens). Note that **dark red** in the figures indicates overlap.

*Table 16.* Effect of calibration dataset size (# of sequences used) on model performance (C4 calibration for 2-bpe quantization).

| # Calib. Seq | DeepSeekV2-Lite | | | Mixtral-8×7B | | |
|---|---|---|---|---|---|---|
| | **WikiText2$^\downarrow$** | **C4$^\downarrow$** | **0-shot$_7^\uparrow$** | **WikiText2$^\downarrow$** | **C4$^\downarrow$** | **0-shot$_7^\uparrow$** |
| 32 | 7.29 | 11.97 | 53.96 | 5.91 | 10.73 | 60.00 |
| 64 | 7.27 | 11.90 | 53.94 | 5.88 | 10.58 | 60.08 |
| 128 | 7.24 | 11.81 | 54.36 | 5.89 | 10.74 | 60.07 |
| 256 | 7.22 | 11.80 | 54.36 | 5.91 | 10.64 | 60.08 |

**Ablation on Extra Constraints in LP Formulation.** We provide additional ablation experiments to evaluate the effect of adding different extra constraints to the LP formulation. Specifically, we compare the following settings on two bit budgets for Mixtral-8×7B bit allocation:

- *w/o Extra Constraints*: Using only the total bit-budget constraint, without any extra constraints;

- *Highest & 2nd Highest* (adopted in the paper): Each layer must be assigned at least one highest-bit and one second-highest-bit expert;

- *Only Highest*: Each layer must be assigned at least one highest-bit expert;

- *Only 2nd Highest*: Each layer must be assigned at least one second-highest-bit expert;

- *Highest Every 2 Layers*: Every two consecutive layers must be assigned at least one highest-bit expert.

As shown in Tab. 17, among all tested variants, enforcing at least one highest-bit and one second-highest-bit expert per layer yields the most favorable results. The benefit of adding this constraint is more pronounced in the 1.5-bit ultra-low-precision setting than in the 2.5-bit setting. We attribute this behavior to the discontinuity of the MoE loss landscape analyzed in Sec. 4.3, where quantization at ultra-low precision can invalidate the loss approximation. Enforcing the high-bit constraint restricts optimization to a mild-error region, which stabilizes the approximation and improves performance.

**Ablation on Expert Bit-width Candidates.** In the main experiments, we use expert bit-width candidates $\{1, 2, 3\}$ and 4-bit attention to align with the prior mixed-precision work PMQ and ensure a fair comparison. Nevertheless, GEMQ is not

*Table 17.* Ablation of adding extra constraints in the global LP formulation on Mixtral-8×7B.

| Constraints | WikiText2$^\downarrow$ | C4$^\downarrow$ | 0-shot$_7^\uparrow$ |
|---|---|---|---|
| | 2.5 bpe | | |
| w/o Extra Constraints | 4.97 | 8.95 | 65.22 |
| Highest & 2nd highest | **4.95** | **8.92** | 65.17 |
| Only Highest | 4.98 | 8.94 | 65.09 |
| Only 2nd highest | 4.97 | 8.94 | **65.24** |
| Highest every 2 layers | 4.97 | 8.95 | 65.22 |
| | 1.5 bpe | | |
| w/o Extra Constraints | 8.81 | 19.42 | 49.80 |
| Highest & 2nd highest | **7.92** | **16.13** | **53.42** |
| Only Highest | 8.02 | 16.87 | 51.31 |
| Only 2nd highest | 8.03 | 17.53 | 50.37 |
| Highest every 2 layers | 8.81 | 19.42 | 49.83 |

restricted to this configuration, as the LP formulation naturally supports arbitrary and richer candidate sets. To demonstrate this flexibility, we provide experiments that expand the expert bit-width candidate set to $\mathcal{B} = \{0, 1, 2, 3, 4\}$ and evaluated additional attention-bit candidates $\{2, 4, 8\}$ on the Mixtral-8×7B model.

As shown in the Tab. 18, expanding the bit-candidate set leads to further improvement in our final ILP optimization objective and the perplexity on the C4 dataset, which is close to our calibration data distribution. These results indicate that our loss approximation and optimization algorithm are working as expected, where better minima can be found with a larger set of feasible solutions. Meanwhile, as a common tradeoff in LLM calibration, improved optimization toward the calibration set slightly hinders generalization performance on other tasks, which is more evident in the 1.5-bit regime. This issue can be resolved by employing a simple mixed-data strategy (as shown in Tab. 3) or exploring generalization objectives like sharpness-aware minimization in future work.

*Table 18.* Ablation of expert bit-width candidates on Mixtral-8×7B (attention bits = 4).

| Bits Per Expert | Bit Candidates $\mathcal{B}$ | Opt Obj (Eq.7)$^\downarrow$ | WT2$^\downarrow$ | C4$^\downarrow$ | 0-shot$_7^\uparrow$ |
|---|---|---|---|---|---|
| | {1,2,3} | 0.0144 | **4.97** | 8.95 | **65.22** |
| 2.5 | {0,1,2,3} | 0.0139 | 5.02 | 8.91 | 64.96 |
| | {1,2,3,4} | 0.0138 | 5.00 | 8.95 | 65.19 |
| | {0,1,2,3,4} | **0.0131** | 5.06 | **8.90** | 65.12 |
| | {1,2,3} | 0.0353 | **8.81** | 19.42 | **49.80** |
| 1.5 | {0,1,2,3} | 0.0314 | 9.41 | 17.34 | 49.28 |
| | {1,2,3,4} | 0.0347 | 9.10 | 17.86 | 49.48 |
| | {0,1,2,3,4} | **0.0308** | 9.65 | **16.85** | 49.35 |

**Ablation on Attention Bit-width.** We also evaluate different attention bit-widths, and the results are shown below. As observed, assigning lower bit-widths (*e.g.*, 2 bits) to the attention modules significantly degrades model performance, which is consistent with prior findings (Kim et al., 2023b; Li et al., 2024) that attention layers are more sensitive to quantization. In contrast, increasing the attention bit-width to 8 bits appears to be a favorable option, as it provides decent performance improvements with only a minimal increase in model size.

*Table 19.* Ablation of attention bit-width (bpe = 2.5, expert bit candidates $\mathcal{B}$ = {1,2,3}).

| Attention Bits | WikiText2$^\downarrow$ | C4$^\downarrow$ | Zero-shot Avg.$^\uparrow$ | Model Size (GB) |
|---|---|---|---|---|
| 8 | 4.81 | 8.63 | 66.44 | 17.00 |
| 4 | 4.97 | 8.95 | 65.22 | 16.37 |
| 2 | 31.14 | 82.79 | 35.16 | 16.06 |

# E. Further Analysis of Global Router Fine-Tuning

**Ablation Study on Router Fine-Tuning Settings.** We ablate the settings for global router fine-tuning in Fig. 7. As shown in the left figure, since routers contain only a small number of parameters, training converges within a single epoch in under 2 minutes. In the right figure, we observe that using more calibration samples can further reduce perplexity on the test set, but the improvement is marginal. We therefore use 128 samples in all experiments.

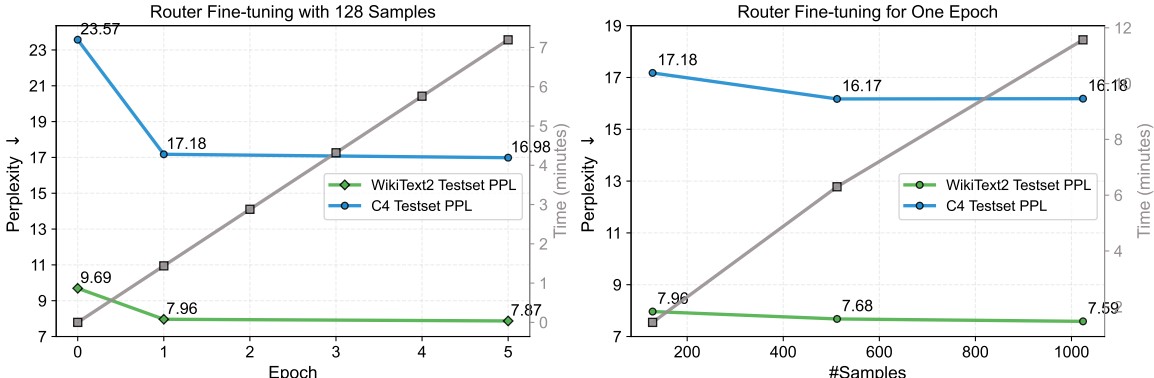

*Figure 7.* Ablation study on router fine-tuning settings. Weights are initialized from the quantized Mixtral-8×7B model (1.5 bits/expert). Training data are randomly extracted from the WikiText2 training set. Times are measured on three H100 GPUs.

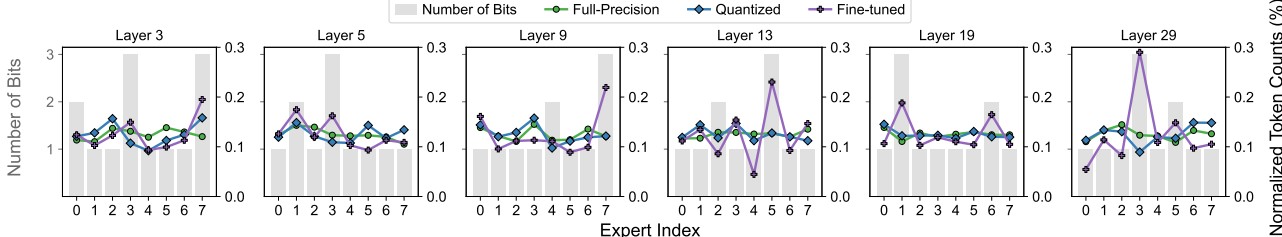

*Figure 8.* Comparison of router statistics from selected layers of the full-precision, quantized (1.5 bits/expert), and router fine-tuned Mixtral-8×7B models on the WikiText2 test set.

**Analysis of Routing Dynamics after Fine-Tuning.** As shown in Fig. 8, several intriguing patterns emerge after fine-tuning. In most cases, the assigned bit-widths of experts exhibit strong correlation with their activation frequencies after fine-tuning, indicating that router fine-tuning increases the selection of higher-bit experts (*e.g.*, layers 3, 5, and 19). This is intuitive as higher-bit experts incur less quantization loss and preserve more information. However, this is not always the case: low-bit experts can also contribute meaningfully to the model. For instance, in layer 9, although expert 4 has a higher bit width, the lower-bit expert 0 is activated more frequently. A potential issue arising from this finding is the frequent use of high-bit experts, which disrupts load balance and increases inference time. However, we observe no noticeable degradation in inference speed compared to the PMQ baseline. In practice, for deployment, high-bit experts can be treated as shared experts and handled separately for further optimization (Dai et al., 2024; Liu et al., 2024a).

Another observation is that the router distribution after fine-tuning differs substantially from that of the FP model, implying that relying on the FP distribution does not lead to optimal selection after expert quantization. This suggests that rigid alignment (Fu et al., 2025; Chen et al., 2025b), which enforces the router to mimic the FP distribution, may be suboptimal for expert-level mixed-precision quantization. Results in Tab. 8 further support this: when we replace the router logits of the quantized model with those from the FP model, the performance remains significantly worse than that of our global router fine-tuning method.

**Router Quantization after Fine-Tuning.** In our method, router weights are stored in 16-bit precision after fine-tuning. Since these weights constitute less than 0.04% of the total parameters, storing them in 16 bits introduces only a negligible increase in model size. However, prior work (Li et al., 2024; Huang et al., 2024a) typically quantizes router weights to 4 bits. For fair comparison, we also report results of quantizing the fine-tuned router weights to 4 bits after fine-tuning in Tab. 20. As shown, quantizing routers to 4 bits has little impact on model performance.

*Table 20.* Perplexity$^{\downarrow}$ of Mixtral-8×7B on WikiText2 and C4 w/ and w/o 4-bit router quantization after fine-tuning.

| #Bits | WikiText2 | | C4 | |
|---|---|---|---|---|
| | 16-bit Routers | 4-bit Routers | 16-bit Routers | 4-bit Routers |
| 2.5 | 5.03 | 5.04 | 9.01 | 9.02 |
| 2.0 | 5.94 | 5.94 | 10.82 | 10.82 |
| 1.5 | 7.71 | 7.72 | 16.85 | 16.82 |

# F. Further Analysis of Progressive Quantization

---

**Algorithm 1:** Progressive Quantization

**Input:** Pretrained FP model $\mathbf{Q}_0$; Calibration datasets; Target bit budgets $\{B_1, \ldots, B_K\}$

Initialize an intermediate quantized model $\mathbf{Q}'$ from FP model $\mathbf{Q}' \leftarrow \mathbf{Q}_0$;

**for** *all $k = 1, 2, \ldots, K$-th target bit budget* **do**

    Collect perturbations $\Delta\mathbf{z}_{ij}$ and layer output gradients $\mathbf{g}_i^{(\mathbf{z})}$ in Eq. 6 from $\mathbf{Q}'$;

    Solve the LP model in Eq. 7 under budget $B_k$ to determine expert bit-width allocation;

    Apply GPTQ pseudo quantization to $\mathbf{Q}_0$ according to the allocation results to obtain $\hat{\mathbf{Q}}_k$;

    Fine-tune routers of the quantized model $\hat{\mathbf{Q}}_k$ to obtain model $\mathbf{Q}_k$;

    Update the intermediate model for progressive quantization $\mathbf{Q}' \leftarrow \mathbf{Q}_k$;

    **if** *Router Quantization is triggered* **then**

        Quantize all routers of $\mathbf{Q}_k$ with GPTQ to the target bit-width;

**return** Quantized models $\{\mathbf{Q}_1, \ldots, \mathbf{Q}_K\}$

---

Algorithm 1 presents the full progressive quantization pipeline, with optional post-quantization router fine-tuning.

**Effectiveness of Progressive Quantization.** Tab. 21 provides ablation results comparing progressive quantization with computing model statistics from the full-precision model. As shown, progressive quantization consistently yields better performance, particularly in the low-bit regime. This finding aligns with our theoretical analysis in Sec. 4.3.

*Table 21.* Ablation study of Progressive Quantization (PQ) on DeepSeekV2-Lite and Mixtral-8×7B. We compare direct quantization from 16-bit (w/o PQ) against our progressive strategy.

| #Bits | Method | DeepSeekV2-Lite | | | Mixtral-8×7B | | |
|---|---|---|---|---|---|---|---|
| | | WT2$^{\downarrow}$ | C4$^{\downarrow}$ | 0-shot$^{\uparrow}$ | WT2$^{\downarrow}$ | C4$^{\downarrow}$ | 0-shot$^{\uparrow}$ |
| 2.5 | w/o PQ | 6.86 | 10.54 | 60.58 | 5.04 | **9.02** | 64.23 |
| | w/ PQ | **6.83** | **10.50** | **60.59** | **5.01** | 9.05 | **64.82** |
| 2.0 | w/o PQ | 7.75 | 12.20 | 53.00 | 6.17 | 11.14 | 58.19 |
| | w/ PQ | **7.74** | **12.12** | **53.42** | **6.09** | **11.00** | **58.60** |
| 1.5 | w/o PQ | 11.72 | 20.87 | 46.17 | 9.21 | 21.12 | 49.24 |
| | w/ PQ | **11.30** | **19.57** | **46.57** | **8.71** | **20.67** | **51.63** |

**Router Change Ratio Comparisons.** Fig. 9 shows the expert change ratio curves, from which the average ratios in Tab. 6 are computed. Each curve shows the router selection change ratio, computed by comparing the 1.5-bpe quantized model with the corresponding model used for expert importance estimation. As seen, using the FP model for 1.5-bit expert importance estimation is inaccurate, as large shifts in token-to-expert assignments induce abrupt loss changes. As we gradually use models with closer bit budgets for estimation, the expert selection change ratio decreases, indicating fewer abrupt loss changes and thus more accurate expert importance estimation. Intriguingly, when the 2-bit fine-tuned model is used for estimation, the resulting 1.5-bit model achieves better perplexity, even though the expert selection change ratio does not improve. We conjecture that this is due to the unstable and rugged loss landscape of the model after fine-tuning, which leads to a relatively high change ratio.

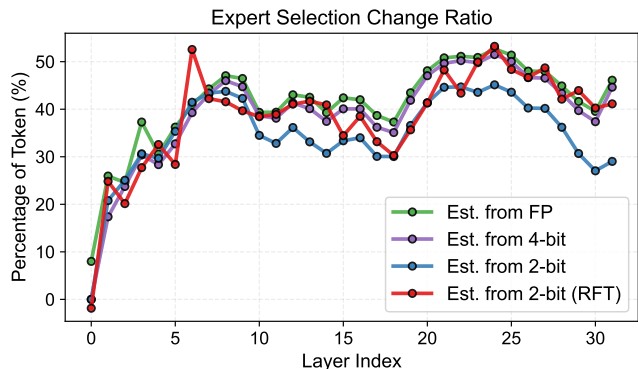

*Figure 9.* Router selection change ratio of Mixtral-8×7B computed on WikiText2 testset.

# G. Detailed Expert Bit-width Allocation Results

Fig. 10 shows detailed expert bit-width allocations produced by GEMQ with candidate set $\mathcal{B} = \{1, 2, 3\}$.

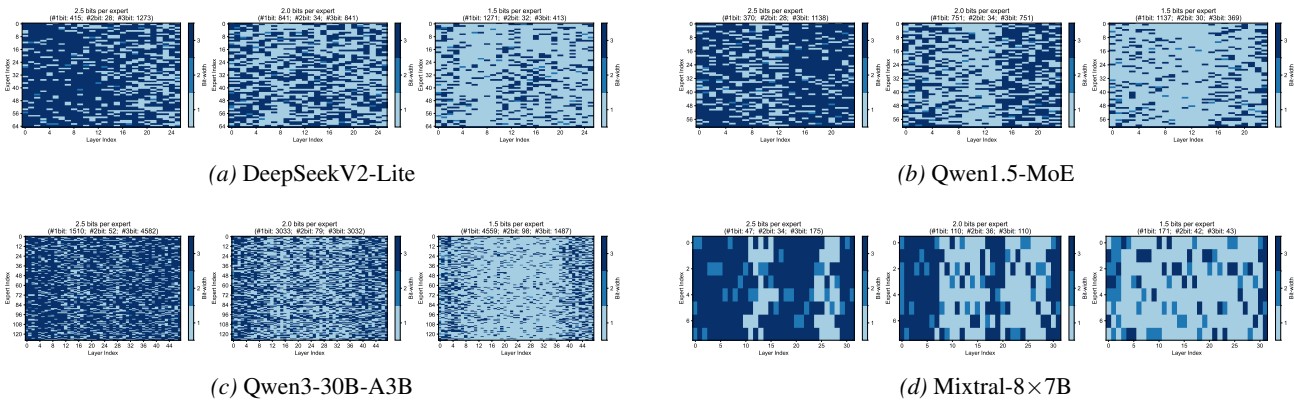

*(a)* DeepSeekV2-Lite

*(b)* Qwen1.5-MoE

*(c)* Qwen3-30B-A3B

*(d)* Mixtral-8×7B

*Figure 10.* Bit-width allocation visualization across different MoE models. Shared experts are merged and positioned last.

