# OpenReview forum: "GEMQ: Global Expert-Level Mixed-Precision Quantization for MoE LLMs"
_ICML.cc/2026/Conference — ICML 2026 regular_

### Official Review · Reviewer_5KAC · 2026-02-23

**Soundness:** 2
**Presentation:** 3
**Significance:** 2
**Originality:** 3
**Overall Recommendation:** 4
**Confidence:** 4

**Summary:**

This paper addresses two key challenges in quantizing MoE-LLMs: (1) existing methods allocate bit budgets only locally within each layer; and (2) low-bit quantization significantly alters router behavior,. To tackle these issues, the authors propose two technical components: (1) a global expert-level linear programming model grounded in task loss and quantization error analysis; and (2) a lightweight PEFT strategy for routers, which aligns pre- and post-quantization routing behavior using a minimal calibration set. Extensive experiments demonstrate that the proposed method significantly reduces memory footprint and accelerates inference with negligible accuracy degradation.

**Compliance With Llm Reviewing Policy:**

Affirmed.

**Final Justification:**

The authors’ experiments and conclusions during rebuttal fully addressed my concerns. Moreover, the empirical results regarding the relationship among quantization overhead, calibration data, and post-quantization model performance align well with my own experience. Given these findings, I believe the paper is now suitable for publication at ICML, and I will maintain my positive recommendation accordingly.

**Key Questions For Authors:**

1. What is the computational overhead introduced by progressive quantization? Does this policy significantly increase the overall quantization time?
2. The router is fine-tuned using only a small calibration set. As mentioned in the weaknesses, how should this set be constructed? Should it aim to cover a diverse range of tokens, or a diverse set of tasks?

**Limitations:**

Yes

**Strengths And Weaknesses:**

__Strengths__:
1. The paper provides a detailed theoretical analysis showing that different layers exhibit different sensitivities to quantization.
2. The presentation is strong, particularly the explanation of why progressive quantization is necessary.

__Weaknesses__:
1. Fine-tuning the router using a small calibration set is no longer a novel idea. While this is not a major issue by itself, the paper does not analyze how the choice of calibration set affects the effectiveness of router fine-tuning.

---

> ### Author Rebuttal · Authors · 2026-03-30
>
> We thank the reviewer for the careful evaluation of our paper and for recognizing its contributions. We have addressed all concerns and questions with additional empirical results.
>
> > **Q1: Computational Overhead of GEMQ Quantization**
>
> We acknowledge that our method incurs higher offline quantization cost than the local method PMQ. To quantify this, we provide **a resource profile including total wall-clock time and peak memory for each quantization stage**. Results are averaged over 3 runs and measured on H100 GPUs.
>
> **Single-Stage Quantization.** As shown for a single quantization round, our method incurs additional memory usage due to gradient computation. However, **the runtime overhead is minimal, accounting for only a small fraction of the total runtime (~6%)**. Importantly, this overhead is a one-time offline cost and is therefore acceptable in practice.
>
> **Progressive Quantization.** When progressively quantizing to lower bit-widths, the runtime overhead accumulates across stages. In practice, we find that a relatively coarse bit-width schedule is sufficient. For example, for 1.5-bit quantization, only three stages (2.5 → 2.0 → 1.5) are needed to achieve strong performance. When the time budget is limited, the number of stages can be further reduced, or gradients and statistics can be computed directly from the full-precision model (i.e., a single stage), at the cost of modest performance degradation.
>
> **Resource Breakdown for Single-Stage Quantization of Mixtral-8x7B (2 BPE)**
> |Resource|Method|Computing Gradients|Computing Stats|Solving LP|Quantization (GPTQ)|Fine-tuning Routers|Total|
> |-|-|:-:|:-:|:-:|:-:|:-:|:-:|
> |**Wall-clock Time (s)**|**PMQ**|N/A|455.6 (21.1%)|0.1 (0.0%)|1706.3 (78.9%)|N/A|2162.0 (100%)|
> ||**GEMQ**|84.3 (3.6%)|468.2 (20.3%)|0.1 (0.0%)|1698.2 (73.5%)|59.4 (2.6%)|2310.2(100%)|
> |**GPU Memory (GiB)**|**PMQ**|N/A|31.8|N/A|21.5|N/A|31.8 (max)|
> ||**GEMQ**|191.2|35.1|N/A|21.5|136.3|191.2 (max)|
>
> **Resource Breakdown for Single-Stage Quantization of DeepSeek-V2-Lite (2 BPE)**
> | Resource | Method | Computing Gradients | Computing Stats | Solving LP | Quantization (GPTQ) | Fine-tuning Routers | Total |
> |-|-|:-:|:-:|:-:|:-:|:-:|:-:|
> | **Wall-clock Time (s)** | **PMQ** | N/A | 721.3 (27.3%) | 0.2 (0.0%) | 1923.7 (72.7%) | N/A | 2645.2 (100%) |
> | | **GEMQ** | 130.5 (4.6%) | 725.3 (25.4%) | 0.2 (0.0%) | 1920.2 (67.3%) | 76.7 (2.7%) | 2852.9 (100%) |
> | **GPU Memory (GiB)** | **PMQ** | N/A | 30.5 | N/A | 8.8 | N/A | 30.5 (max) |
> | | **GEMQ** | 82.1 | 32.8 | N/A | 8.8 | 55.2 | 82.1 (max) |
>
> > **W1 & Q2: Construction of Calibration Set for Router Fine-tuning**
>
> We thank the reviewer for raising this important question. We agree that the composition of the calibration set for router fine-tuning warrants further analysis.
> To investigate this, we conduct ablation experiments on Mixtral-8×7B (2 BPE) using different calibration datasets for router fine-tuning, while keeping the number of calibration samples fixed. The results are summarized below:
>
> | Calibration Data | WT2 PPL$^\downarrow$ | C4 PPL$^\downarrow$ | Zero-Shot ACC$^\uparrow$ |
> | :--- | :-: | :-: | :-: |
> | None (w/o R-FT) | 6.12 | 11.08 | 59.45 |
> | WT2 | 6.03 | 10.89 | 59.98 |
> | C4 | 5.96 | 10.45 | 60.26 |
> | MATH | 6.00 | 10.70 | 60.25 |
> | MATH+C4 | 6.02 | 10.75 | 60.89 |
>
> From these results, we make the following observations:
>
> * **Router fine-tuning consistently improves performance** across both language modeling (WT2/C4 perplexity) and downstream tasks (zero-shot accuracy), regardless of the calibration dataset used.
> * **The performance differences across calibration datasets are relatively small**, suggesting that the method is robust to the specific choice of calibration data.
> * Among different choices, **C4 provides a strong general-purpose baseline**, while **incorporating domain-specific data can further improve downstream performance.**
>
> These findings suggest that when the target application is unknown, a general-purpose dataset such as C4 provides a reasonable balance across tasks (e.g., general QA and reasoning). When the target domain is known (e.g., mathematical reasoning), incorporating domain-specific data can further improve performance.
> We note that the observations above are consistent with recent findings on calibration data curation [1], which highlight that effective calibration data should provide diverse and representative coverage of data patterns. We believe [1] can serve as a useful guideline for selecting calibration data for router fine-tuning.
>
> **Overall, our results suggest that calibration data should be both diverse and aligned with the target tasks, while our method remains robust under simple calibration choices.** We will include additional discussion of these findings in the revision and leave a more systematic study of calibration data construction to future work.
>
> [1] He et al., "Preserving LLM Capabilities through Calibration Data Curation.", NeurIPS (2025)

---

> > ### Author Rebuttal · Reviewer_5KAC · 2026-04-01
> >
> > The authors’ experiments and conclusions address my exact concerns. Moreover, the empirical results regarding the relationship among quantization overhead, calibration data, and post-quantization model performance align well with my own experience. Given these findings, I believe the paper is now suitable for publication at ICML, and I will maintain my positive recommendation accordingly.

---

> > > ### Author Response · Authors · 2026-04-02
> > >
> > > Thank you for the positive feedback and for maintaining your positive recommendation. We are very pleased to hear that the additional experiments and clarifications have fully addressed your concerns.
> > >
> > > We will incorporate these findings and the corresponding discussions into the final revision to further improve the clarity and completeness of the paper.

---

### Official Review · Reviewer_yuuH · 2026-02-26

**Soundness:** 3
**Presentation:** 3
**Significance:** 3
**Originality:** 3
**Overall Recommendation:** 4
**Confidence:** 3

**Summary:**

The paper addresses the significant memory bottleneck of deploying Mixture-of-Experts Large Language Models (MoE-LLMs) by proposing GEMQ, a Global Expert-level Mixed-precision Quantization framework. The authors identify two major flaws in existing MoE quantization methods: they allocate bits locally (layer-by-layer) and ignore the shifts in token-to-expert routing caused by the quantization of expert weights. To solve this, GEMQ introduces a global linear programming (LP) formulation based on error analysis (using the Fisher Information Matrix to approximate the Hessian) to allocate bit-widths across all experts model-wide. Furthermore, it employs a highly efficient global router fine-tuning step to realign the routing dynamics with the quantized experts. These techniques are embedded in a progressive quantization pipeline.

**Compliance With Llm Reviewing Policy:**

Affirmed.

**Final Justification:**

The authors have successfully addressed my concerns regarding calibration sensitivity, computational overhead, and theoretical assumptions. The proposed GEMQ framework offers a solid and practical contribution to MoE quantization. Therefore, I maintain my score.

**Key Questions For Authors:**

1. What is the precise end-to-end wall-clock time overhead of the entire Progressive Quantization pipeline (e.g., for Mixtral-8x7B) compared to a standard one-shot PTQ baseline like PMQ?

2. Given the high sensitivity to calibration data distributions (Table 3), how should practitioners construct a "safe" calibration set without prior knowledge of downstream use cases? Are there regularization terms that could be added to the global LP to prevent over-fitting to the calibration domain's Fisher Information Matrix?

3. In Figure 9, the router selection change ratio remains quite high (around 30-40%) even with progressive quantization and router fine-tuning at 1.5-bits. Does this suggest that updating only the routers is insufficient to fully correct the representation shift at extreme low bits?

**Limitations:**

Yes

**Strengths And Weaknesses:**

Strengths:
1. Identifying that expert quantization severely distorts the router distribution (e.g., over 40% of tokens changing experts under 1.5-bit quantization) is a keen and crucial observation. Addressing this via parameter-efficient router fine-tuning is an elegant, lightweight, and highly effective solution.
2. Transitioning from heuristic layer-wise bit allocation to a global LP formulation grounded in error analysis is a natural and mathematically sound progression for the field. It effectively exploits the heterogeneous importance of layers and experts.

Weaknesses:
1. As shown in the GSM8K results (Table 3), the global LP formulation is highly sensitive to the calibration data. Relying purely on general text (C4) causes a sharp drop in math reasoning, necessitating a manually curated mixed dataset (M+C4). This raises concerns about the method's robustness and generalization when practitioners lack prior knowledge of the target downstream tasks.
2. The Progressive Quantization (PQ) framework requires an iterative loop of "error estimation $\rightarrow$ LP allocation $\rightarrow$ quantization $\rightarrow$ router fine-tuning". While the authors note that a single router fine-tuning epoch takes only a few minutes, the total end-to-end wall-clock time of the entire PQ pipeline compared to one-shot baselines (like PMQ) is not clearly presented.
3. The error approximation relies on the assumption that the pre-trained model is at a local minimum (zero gradient). Even with progressive quantization attempting to smooth the transition, injecting massive quantization noise at 1.5-bits inevitably violates this Taylor expansion assumption, making the theoretical foundation slightly fragile at the extremes.

---

> ### Author Rebuttal · Authors · 2026-03-30
>
> We thank the reviewer for the insightful feedback on the theoretical and practical robustness of our framework. We have addressed all concerns with additional experiments below.
>
> > **W1 & Q2: Sensitivity to Calibration Data Distribution**
>
> Thank you for raising this practical question.
> A key observation from our study is that **the calibration dataset should be diverse and exhibit domain correspondence**, i.e., it should align with the target deployment domain. This is also consistent with recent findings on calibration data curation [1].
> In practice, when the target application is unknown, **using a general-purpose dataset such as C4 provides a reasonable balance across tasks** (e.g., general QA and reasoning). When the target domain is known (e.g., mathematical reasoning), **incorporating domain-specific samples into the calibration set can further improve performance**.
>
> We agree that explicit regularization can mitigate domain overfitting, but it may also hinder alignment with the underlying data distribution. Instead, we emphasize constructing a diverse and representative calibration set (e.g., following [1]). A sufficiently diverse dataset  that spans a wide range of activation patterns acts as an implicit form of regularization, improving robustness while reducing overfitting to narrow domains.
>
> [1] He et al., "Preserving LLM Capabilities through Calibration Data Curation.", NeurIPS (2025)
>
> > **W2 & Q1: Computational Overhead of GEMQ Quantization**
>
> We conduct a resource breakdown analysis to quantify the quantization overhead introduced by GEMQ. Please refer to our response to `Reviewer 5KAC` for detailed results. **In summary, GEMQ incurs only a minimal one-time offline overhead (less than 3 minutes), making it practical for deployment.**
>
> > **W3: Validity of the Theoretical Assumption at Low Bit-Widths**
>
> Thank you for the insightful observation. Injecting quantization noise, especially at extremely low-bit settings, initially perturbs the model away from a local minimum. However, **we empirically show that router fine-tuning effectively brings the model back to a local minimum**, restoring the validity of the first-order gradient assumption for the next progressive quantization step.
>
> We validate this through two observations:
> (1) **Convergence behavior.** The model converges after router fine-tuning, with stable perplexity observed as shown in Figure 7;
> (2) **Gradient magnitude.** The average gradient magnitude across all experts becomes significantly smaller after fine-tuning compared to the quantized model, indicating that the model returns to a local minimum regime.
>
> |Mixtral-8x7B|FP16|Quantized (1.5 BPE)|Fine-tuned (1.5 BPE)|
> |-|:-:|:-:|:-:|
> |Average Gradient Magnitude|1.393|1.626|1.410|
>
> **Worst-case Analysis.**
> To assess the impact when this assumption breaks down at extremely low bit-widths, we report in Table 5 a comparison of expert importance estimated under different settings. As shown, computing expert importance directly from the quantized model increases perplexity by at most 1.0, indicating that **global bit allocation remains robust without catastrophic failure** and incurs only modest degradation.
>
> > **Q3: Router Selection Change Ratio in Extreme Low Bits**
>
> Thank you for the insightful observation. We agree that router-only fine-tuning cannot fully recover performance at extremely low bit-widths. It is designed to **strike a balance between performance and efficiency**: remaining lightweight in terms of data and compute, while mitigating performance degradation caused by routing distortion. Furthermore, progressive quantization, combined with router fine-tuning, improves the accuracy of expert importance estimation and enhances overall performance.
>
> In contrast, full-model quantization-aware training (i.e., jointly tuning both the router and expert weights) can further improve performance in extreme low-bit settings, but incurs substantially higher computational cost, making it less suitable for MoE LLMs.

---

> > ### Author Rebuttal · Reviewer_yuuH · 2026-04-03
> >
> > Thank you for the detailed and well-structured rebuttal. All concerns have been satisfactorily addressed.
> >
> > I am satisfied with the rebuttal and will maintain my score of 4 (Weak Accept). The paper makes a solid and practical contribution to MoE quantization.

---

> > > ### Author Response · Authors · 2026-04-04
> > >
> > > Thank you for the positive feedback and for recognizing our contribution. We are glad that our responses have satisfactorily addressed your concerns. We will incorporate the clarifications and additional analysis into the final revision to further strengthen the paper.

---

### Official Review · Reviewer_Xtdy · 2026-03-11

**Soundness:** 2
**Presentation:** 3
**Significance:** 2
**Originality:** 2
**Overall Recommendation:** 3
**Confidence:** 4

**Summary:**

This paper proposes a expert-wise mixed-precision quantization approach for MoE. Instead of allocating precisions locally layer-by-layer, the paper formulates a method based on the approximation of the loss-increment due to quantization to estimate the global importance of the experts. Furthermore, the paper proposes to fine-tune the routers after expert quantization for improved performance. Finally, the paper proposes a progressive quantization strategy to extremely low bits, where instead of estimating experts' importance based on the full-precision model, the importance are estimated based on previously quantized and fine-tuned model. The paper provides experiments on multiple standard MoE models for benchmark tasks to support their claim.

**Compliance With Llm Reviewing Policy:**

Affirmed.

**Final Justification:**

The authors tried to address the weaknesses mentioned related to the computational overhead of progressive bit-allocation and block-wise approximation by arguing that:

1. It is worthwhile to spend extra one-time compute to gain better performance

2. It is a common practice to adopt block-wise approximation in PTQ literature

However, the first argument raises question that whether the improvement is merely due to the allocation of extra compute or technical innovation. The compute overhead will grow with model size (e.g., in my understanding it will be large for huge size MoE models such as DeepSeek-R1. Indeed, SOTA MoE models are huge). In that case, the question remains open about rather spending compute in other compute-heavy processes such as distillation. Moreover, despite the compute overhead the method relatively underperforms in larger size model presented in the paper i.e., Mixtral 8x7B.

Additionally, the second argument undermines novelty of this work.

Overall, due to the remaining open questions I retain my original score.

**Key Questions For Authors:**

See weaknesses

**Limitations:**

Yes

**Strengths And Weaknesses:**

**Strengths**

1. The proposed global importance estimation is novel and intuitive
2. The fine-tuning of the routers after quantization and the performance improvements is promising
3. The paper is well-written and easy to follow

**Weaknesses**

1. The iterative approach and the approximation of the loss-increment through gradient can be prohibitively time consuming and computationally expensive. The author did not provide any comparison of time and computational resources required for bit allocation between the proposed method and local layer-wise strategy

2. The uniform allocation of average bit-width per expert in each layer allows homogeneity, which may allow to realize a higher throughput. The author did not provide any comparison of the speed between the proposed method and the local-strategy.

3. The approximation of equation (3) by equation (4) is based on the assumption that the weights outside the block are not perturbed. Therefore, equation (6) should be based on the condition that, other experts are not quantized. However, the approximation can be severely exacerbated when the quantization error of one expert will affect the other.

4. It’s not clear why updating only the routers improve the performance. Do the updated routers recover to the initial state before quantization in terms of routing decisions? No theoretical or empirical analysis is provided.

5. Several inconsistencies in the experimental results:

    (i)  How can you achieve 2.5 bpe and 1.5 bpe for uniform case? If you allocate the same bit from {1,2,3} to all experts, how the average bpe is a fraction?

    (ii) In Table 2, row 2 (GEMQ) and row 9 (GEMQ) showing different results despite having same quantization configuration (2.5-16).

6. What does “squared gradients” refer to in Fig. 1?

---

> ### Author Rebuttal · Authors · 2026-03-30
>
> We sincerely thank the reviewer for the detailed review and valuable feedback. We have carefully addressed all concerns by clarifying misunderstandings and providing additional empirical results.
>
> > **W1: Computational Overhead of GEMQ Quantization**
>
> We conduct a resource breakdown analysis to quantify the quantization overhead introduced by GEMQ. Please refer to our response to `Reviewer 5KAC` for detailed results. **In summary, GEMQ incurs only a minimal one-time offline overhead (less than 3 minutes), making it practical for deployment.**
>
> > **W2: Throughput Comparison against Uniform Baselines**
>
> Thank you for the thoughtful question. We compare against the 2-bit Uniform and PMQ baselines and observe that **GEMQ achieves comparable decoding throughput**. This is because our fused MoE kernel supports mixed-precision (1/2/3-bit) dequantization with grouped GEMM, minimizing the overhead of switching across experts with different bit-widths. Please refer to Table 13 for additional comparisons across different bit-widths and models.
>
> | Mixtral-8x7B (2 BPE) | Peak Memory (GiB) | Throughput (tokens/s) |
> | :--- | :-: | :-:|
> | Uniform | 14.9 |85.8 |
> | PMQ | 14.9 | 84.9 |
> | GEMQ | 14.9 | 84.6 |
>
> > **W3: Block-wise Approximation**
>
> We agree that our analysis assumes block-wise independence, whereas in practice quantization errors may propagate across layers and affect subsequent experts. Nevertheless, we note that this assumption is commonly adopted in existing PTQ literature, as it provides a practical trade-off: accurately modeling inter-layer interactions in large-scale MoE models is computationally intractable, while a block-wise formulation offers a tractable and empirically effective approximation.
>
> > **W4: Effectiveness of Router Fine-Tuning**
>
> Thank you for the insightful question. Quantization perturbs both expert behaviors and router inputs, making the original routing decisions suboptimal. Router fine-tuing resolves this mismatch by adapting the routing policy to the quantized experts. The gains mainly stem from two aspects: (i) **improved expert selection under perturbation** (Table 4) and (ii) **more accurate estimation of expert importance** (see Figure 3 for theoretical analysis and Table 5 for empirical results). Importantly, **the router does not recover pre-quantization decisions but instead learns a new strategy tailored to the quantized model** (see Appendix E for analysis). Together, these empirical results show that removing router fine-tuning leads to significant performance degradation.
>
> > **W5: Experimental Settings**
>
> We apologize for the confusion. We clarify that: (i) to achieve average BPEs of 2.5/1.5 in the Uniform setting, we follow semi-uniform configurations from prior work [1,2], applying 3/2-bit quantization to the first half of the layers and 2/1-bit to the second half; (ii) the discrepancy in zero-shot results is due to different evaluation task sets, as noted in Table 2 ("specific task sets"). Detailed results and configurations are provided in Tables 10–12 and Appendix B.
>
> [1] Li et al., "QuantMoE-Bench: Examining Post-Training Quantization for Mixture-of-Experts." arXiv preprint arXiv:2406.08155 (2024).
> [2] Chen et al., "EAC-MoE: Expert-selection aware compressor for mixture-of-experts large language models." ACL (2025).
>
> > **W6: Definition of "Squared Gradients"**
>
> We apologize for the confusion. The term "squared gradients" refers to the trace of the empirical Fisher Information Matrix of the expert weights, which we use as a proxy for Hessian-based sensitivity of each expert. Specifically, for a gradient vector $\mathbf{g} \in \mathbb{R}^N$ of expert weights $\mathbf{w} \in \mathbb{R}^N$, this corresponds to $\sum_{i=1}^N g_i^2$, where $N$ is the number of parameters in the expert. We will clarify this in Figure 1 to avoid ambiguity.

---

> > ### Author Rebuttal · Reviewer_Xtdy · 2026-04-03
> >
> > 1. Computational overhead for progressive quantization is a fundamental weakness of this work
> >
> > 2. Block-wise approximation is still a weakness.
> >
> > 3. I am still not convinced with the intuition behind the router's update. In fact, the authors response raised more concerns. Specifically, the claim that "the router does not recover pre-quantization decisions but instead learns a new strategy tailored to the quantized model". Does it mean that the same token now requires a different intermediate representation through a different expert compared unquantized model?

---

> > > ### Author Response · Authors · 2026-04-04
> > >
> > > Thank you for the follow-up questions. We appreciate the opportunity to further clarify these important points.
> > >
> > > > **W1: Computational overhead is a fundamental weakness**
> > >
> > > We respectfully disagree that the overhead of progressive quantization constitutes a **fundamental weakness**. Quantization is designed to improve **inference-time efficiency**. It is standard practice to incur a small one-time offline quantization cost to achieve substantial inference gains. Importantly, our method introduces **no additional overhead during inference**, and therefore does not affect deployment efficiency.
> > >
> > > Regarding the one-time offline overhead, we have shown that it is comparable to existing methods and represents a **configurable design trade-off**. We further clarify below.
> > >
> > > **Single-stage Overhead**. The core components of GEMQ (gradient computation and router fine-tuning) introduce **less than 3 minutes per stage** compared to PMQ on large MoE models.
> > >
> > > **Configurable Design.** The multi-stage progressive quantization is a flexible design that enables a trade-off between accuracy and overhead. In practice, three stages (≈2 hours in total) are sufficient to achieve robust performance at 1.5-bit. **This one-time cost is negligible relative to the sustained inference efficiency gains achieved after quantization**.
> > >
> > > When time is constrained, the number of stages can be reduced accordingly. Notably, even in the extreme case of a single-stage variant (i.e., without progressive quantization), our method still outperforms PMQ with nearly the same runtime, as shown below.
> > >
> > > |Mixtral-8x7B (1.5 BPE)|WT2 PPL|C4 PPL|Quantization Time (hours)
> > > |-|:-:|:-:|:-:|
> > > |PMQ|8.47|20.77|0.60|
> > > |GEMQ (1 stage)|8.11|17.16|0.64|
> > > |GEMQ (3 stages)|7.93|16.20|1.93|
> > >
> > > > **W2: Block-wise approximation is still a weakness**
> > >
> > > We note that the block-wise assumption follows common practice in well-established post-training quantization methods [1–5].
> > > In addition, other reviewers also consider this formulation reasonable. `Reviewer TNuq` states that "the core assumptions are reasonable and the derivation process is standardized", and `Reviewer yuuH` comments that "global LP formulation is a natural and mathematically sound progression."
> > >
> > > Beyond these, we provide more empirical evidence on the effectiveness of the block-wise error approximation. Specifically, we conduct a **correlation study** between (i) the total loss increase predicted by summing expert-wise estimates from our method and (ii) the true loss increase after joint quantization, to evaluate its reliability in estimating the overall loss.
> > >
> > > We perform this study on Mixtral-8x7B across 23 quantization configurations with varying bit-widths, covering a diverse set of regimes. The results show **strong agreement between the predicted and true loss increase** (Pearson = 0.98, Spearman = 0.96, Kendall’s tau = 0.86).
> > >
> > > These findings indicate that, although expert interactions are not explicitly modeled, **the block-wise formulation provides a faithful estimate of the overall loss increase in practice**.
> > >
> > > [1] Up or Down? Adaptive Rounding for Post-Training Quantization. ICML (2020).
> > > [2] HAWQ-V2: Hessian Aware trace-Weighted Quantization of Neural Networks. NeurIPS (2020).
> > > [3] Accurate Post Training Quantization With Small Calibration Sets. ICML (2021).
> > > [4] BRECQ: Pushing the Limit of Post-Training Quantization by Block Reconstruction. ICLR (2021).
> > > [5] SqueezeLLM: Dense-and-Sparse Quantization. ICML (2024).
> > >
> > > > **W3: Intuition behind the router's update**
> > >
> > > We understand the reviewer’s concern as whether routing decisions should remain consistent with the full-precision model and whether deviations may affect token semantics.
> > >
> > > This is not the case. Quantizing experts introduces two perturbations: (i) **input shift** from accumulated quantization noise, and (ii) **expert capability shift** due to mixed-precision expert quantization. As a result, **the original routing decisions are no longer optimal for the quantized model**. Router fine-tuning therefore serves to **re-align routing with the modified experts**, ensuring effective processing under these perturbations.
> > >
> > > Importantly, routing a token to a different expert does **not imply that its semantic representation is altered**, because of the inherent **functional redundancy** among MoE experts, which allows routing to shift to a functionally similar but less affected expert to minimize the final cross-entropy loss.
> > >
> > > We verify this via ablation. Enforcing full-precision (FP) routing improves over the unadapted router, but remains inferior to fine-tuning, indicating that **preserving original routing is suboptimal, while adapting routing achieves better performance**.
> > >
> > > |Mixtral-8x7B (1.5 BPE)|WT2 PPL|C4 PPL
> > > |-|:-:|:-:|
> > > |w/o Router Adaptation|9.66|23.53
> > > |w/ FP Router Decisions|9.29|22.72
> > > |w/ Router Fine-Tuned|7.69|17.18
> > >
> > > Additionally, router fine-tuning improves expert importance estimation during progressive quantization (Section 4.2 & Table 5).

---

### Official Review · Reviewer_TNuq · 2026-03-16

**Soundness:** 3
**Presentation:** 4
**Significance:** 3
**Originality:** 3
**Overall Recommendation:** 5
**Confidence:** 4

**Summary:**

This paper focuses on the core memory bottleneck problem in the deployment of hybrid expert large models. Addressing two major shortcomings of existing MoE mixed-precision quantization methods: first, intra-layer local bit allocation ignores the differences in expert importance between layers; second, quantization-induced routing distribution offset leads to suboptimal routing decisions. This paper proposes GEMQ, a global expert-level hybrid-precision quantization framework.

The core design of this framework comprises three organically integrated modules: First, a global linear programming (LP) bit allocation model is constructed based on quantization error analysis. This model achieves cross-layer comparable expert importance modeling based on task loss, extending traditional intra-layer local LP to global optimization of the entire model, and using heuristic coefficients that do not require manual parameter tuning. Second, a parameter-efficient global routing fine-tuning strategy fine-tunes routing weights, accounting for less than 0.04% of the total parameters, allowing the routing mechanism to adapt to the quantized expert characteristics, solving the routing offset problem caused by quantization, with an additional time overhead of less than 5%. Third, a progressive quantization process iteratively optimizes expert importance estimation through the model fine-tuned in the previous stage, alleviating the Taylor approximation failure problem of the full-precision model in low-bit scenarios and further improving quantization performance in extremely low-bit scenarios.

**Compliance With Llm Reviewing Policy:**

Affirmed.

**Final Justification:**

I would like to thank the authors for their extremely thorough and constructive rebuttal. The authors have directly addressed all of my concerns with substantial new experiments and convincing quantitative analysis. I am very satisfied with the response.

**Key Questions For Authors:**

1. The paper's quantization error analysis and expert importance estimation are based on the core assumption that the model converges to a local minimum and the first-order gradient term is negligible. However, the model after low-bit quantization and routing fine-tuning may deviate from this local minimum. Have the authors verified, through theoretical analysis or experiments, that this assumption still holds true in the quantized/fine-tuned model? If this assumption fails in extremely low-bit scenarios, to what extent will it negatively impact the expert importance estimation of global bit allocation?
2. The paper observes that after route fine-tuning, the model tends to activate high-bit experts more, while stating that no decrease in inference speed was observed, but it does not provide a quantitative indicator of expert load balancing. In extremely low-bit scenarios such as 1.5 bits, will route fine-tuning lead to an imbalance in expert load? If this problem exists, will it be amplified in long-sequence, large-batch inference scenarios?

**Limitations:**

The authors have not yet directly discussed the limitations.

The content of this paper is currently relatively complete. However, it is still recommended that the author supplement some information regarding the limitations of the work to comprehensively improve the overall quality of the article. Specific suggestions for improvement can refer to the Strengths and Weaknesses and Key Questions sections.

**Strengths And Weaknesses:**

1. Strengths
1.1 This paper derives a surrogate index for expert importance based on Taylor expansion and quantization error analysis, and uses a diagonal Fisher information matrix to approximate the Hessian. The core assumptions are reasonable and the derivation process is standardized. Furthermore, the paper validates the model on four MoE models with different characteristics, demonstrating comprehensive experimental coverage.
1.2 The diagrams in this paper are intuitive and well-organized. Motivation Figure 1 clearly illustrates the heterogeneity of inter-layer expert importance and the routing offset problem after quantization, and fully presents the entire process of progressive quantization. The paper provides all the details, including model configuration, and the appendix supplements the complete algorithm flow, additional experimental results, and ablation analysis.
1.3 The memory bottleneck of the MoE architecture is a core obstacle to its large-scale industrial deployment. The method proposed in this paper can retain the core capabilities of the model even at extremely low bit depths, reducing the deployment threshold of mainstream MoE models such as Mixtral-8×7B from high-end multi-GPU clusters to a single consumer-grade GPU. Furthermore, the method is not dependent on a specific MoE architecture design and performs excellently on MoE models with different parameter levels and expert configurations.

2. Weaknesses
2.1 All existing experiments are based on pre-trained pedestal models and have not been validated on instruction fine-tuning and human-aligned dialogue models widely used in industry. At the same time, the experiments only cover sequences of length 2048 and have not verified the cumulative effect of quantization error and changes in routing behavior in long text scenarios (32k+).
2.2 The paper observes that after routing fine-tuning, the model tends to activate high-bit experts more, which only indicates that no decrease in inference speed was observed. However, supplementing the analysis with quantitative analysis of the core indicators of expert load balancing (such as the coefficient of variation of expert activation frequency and load balancing loss) can provide a more comprehensive analysis, including the potential impact on inference performance.

---

> ### Author Rebuttal · Authors · 2026-03-30
>
> We thank the reviewer for the constructive feedback and insightful questions. We have addressed all concerns with additional empirical results.
>
> > **W1: Validation on Instruction Models and Long-Context Scenarios**
>
> We completely agree that evaluating on instruction-tuned models and long-context scenarios is crucial. Following your suggestion, we have substantially expanded our experiments to cover both.
>
> **Instruction Models.** We note that the main paper reports results on the post-trained model Qwen3-30B-A3B. Here, we additionally evaluate an instruction-tuned model, Mixtral-8×7B-Instruct. As shown, GEMQ consistently outperforms Uniform and PMQ across benchmarks, demonstrating effectiveness on human-aligned dialogue models.
>
> |Mixtral-8x7B-Instruct|WT2 PPL$^\downarrow$|C4 PPL$^\downarrow$|Zero-Shot ACC$^\uparrow$|
> |-|:-:|:-:|:-:|
> |FP16|4.14|7.78|72.95|
> |Uniform (2 BPE)|6.29|11.47|62.10|
> |PMQ (2 BPE)|6.09|11.05|63.97|
> |GEMQ (2 BPE)|**6.01**|**10.96**|**64.17**|
>
> **Long-context Evaluation.** We further evaluate the quantized Mixtral-8×7B on the NIAH subtasks from the long-context benchmark RULER [1], across context lengths ranging from 8K to 32K tokens. The table below reports retrieval accuracy, showing that our method consistently outperforms the baseline while preserving long-context capabilities even at extremely low bit-widths.
>
> |Mixtral-8x7B|8K$^\uparrow$|16K$^\uparrow$|32K$^\uparrow$|
> |-|:-:|:-:|:-:|
> |FP16|1.00|0.97|0.94|
> |PMQ (2 BPE)|0.89|0.84|0.72|
> |GEMQ (2 BPE)|**0.93**|**0.87**|**0.77**|
>
> [1] Hsieh et al., "RULER: What's the real context size of your long-context language models?." arXiv preprint arXiv:2404.06654 (2024).
>
> > **W2 & Q2: Quantitative Load Balancing Analysis**
>
> We agree that a more explicit analysis of expert load balancing would strengthen the paper. To quantify potential imbalance, we report (1) the coefficient of variation of expert activation frequencies, (2) the auxiliary load-balancing loss, and (3) the minimum expert activation frequency, comparing the full-precision model with our quantized, fine-tuned models on the WikiText2.
>
> The results below show that the quantized model introduces only a **moderate increase in activation variance** compared to the FP model. Importantly, although higher-bit experts are selected more frequently, **the overall activation distribution remains well spread**: all experts continue to receive non-trivial routing probability, and no expert is consistently inactive, indicating that **no expert collapse occurs**. This behavior is also consistent with the activation frequency patterns shown in Figure 8.
>
> |Mixtral-8x7B|Coefficient of Variation|Load-balancing Loss|Minimum Activation Frequency|
> |:- |:-:|:-:|:-:|
> |FP16|0.139|2.002|0.028|
> |GEMQ (1.5 BPE)|0.197|2.003|0.032|
>
> **Impact on Inference Efficiency.**
> Although higher-bit experts are selected slightly more often, this **does not translate into significant slowdown** (less than 3 tokens/s decrease in throutput compared with a uniform baseline).
> This is because our fused MoE kernel supports mixed-precision (1/2/3-bit) dequantization with grouped GEMM, reducing the overhead of switching across experts with different bit-widths.
> We further evaluate decoding latency at a longer sequence length of 4K. Consistent with Table 13, **our method achieves latency comparable to the Uniform and PMQ baselines**, confirming its practicality for long-sequence inference.
>
> > **Q1: Validity of the Local Minimum Assumption Post-Quantization**
>
> We thank the reviewer for the insightful question. Injecting quantization noise, especially at extremely low-bit settings, initially perturbs the model away from a local minimum. However, **we empirically show that router fine-tuning effectively brings the model back to a local minimum**, restoring the validity of the first-order gradient assumption for the next progressive quantization step.
>
> We validate this through two observations:
> (1) **Convergence behavior.** The model converges after router fine-tuning, with stable perplexity observed as shown in Figure 7;
> (2) **Gradient magnitude.** The average gradient magnitude across all experts becomes significantly smaller after fine-tuning compared to the quantized model, indicating that the model returns to a local minimum regime.
>
> |Mixtral-8x7B|FP16|Quantized (1.5 BPE)|Fine-tuned (1.5 BPE)|
> |:-|:-:|:-:|:-:|
> |Average Gradient Magnitude|1.393|1.626|1.410|
>
> **Worst-case Analysis.**
> To assess the impact when this assumption breaks down at extremely low bit-widths, we report in Table 5 a comparison of expert importance estimated under different settings. As shown, computing expert importance directly from the quantized model increases perplexity by at most 1.0, indicating that **global bit allocation remains robust without catastrophic failure** and incurs only modest degradation.
>
> > **L1: Limitations Section**
>
> We appreciate the suggestion and will add a dedicated Limitations section to the paper.

---

### Decision · Program_Chairs · 2026-04-30

**Decision:**

Accept (regular)

**Comment:**

This paper explores techniques to improve the mixed-precision quantization of MoE-LLMs. Specifically, it proposes a global linear programming–based bit allocation model to replace prior layer-by-layer bit allocation, and introduces router fine-tuning to adapt routing to quantized experts. The methods are evaluated on several popular MoE-LLMs, showing improved low-precision accuracy over prior baselines.

The paper is well written and easy to follow. The two major issues in existing MoE quantization methods—layer-by-layer bit allocation and router shift induced by quantization—are well analyzed with strong supporting evidence. The proposed methods are both novel and intuitive. The evaluation is thorough across a range of MoE-LLM families and demonstrates performance gains over strong baselines.

During rebuttal, the authors provided additional results on instruction-tuned models, long-context evaluation, load balancing analysis, calibration dataset sensitivity, as well as the computational overhead of the quantization process. These additions clarified most of the reviewers’ concerns and further strengthened the paper.

There remains a concern regarding the relatively expensive offline quantization process; however, overall, the paper makes a solid and practical contribution to MoE-LLM quantization. Therefore, the paper is recommended for acceptance.